# Ligand efficacy modulates conformational dynamics of the μ-opioid receptor

Jiawei Zhao[1,2,3,9], Matthias Elgeti[4,5,9 ✉], Evan S. O'Brien[6], Cecília P. Sár[7], Amal El Daibani[8], Jie Heng[1,2], Xiaoou Sun[1,2], Elizabeth White[6], Tao Che[8], Wayne L. Hubbell[4], Brian K. Kobilka[6 ✉] & Chunlai Chen[1,3 ✉]

The μ-opioid receptor (μOR) is an important target for pain management[1] and molecular understanding of drug action on μOR will facilitate the development of better therapeutics. Here we show, using double electron–electron resonance and single-molecule fluorescence resonance energy transfer, how ligand-specific conformational changes of μOR translate into a broad range of intrinsic efficacies at the transducer level. We identify several conformations of the cytoplasmic face of the receptor that interconvert on different timescales, including a pre-activated conformation that is capable of G-protein binding, and a fully activated conformation that markedly reduces GDP affinity within the ternary complex. Interaction of β-arrestin-1 with the μOR core binding site appears less specific and occurs with much lower affinity than binding of G_i.

μOR is a family A G-protein-coupled receptor (GPCR) and an important drug target for analgesia. However, activation of the μOR by opioids such as morphine and fentanyl may also lead to adverse effects with varying severity, including constipation, tolerance and respiratory depression. The μOR activates G_{i/o} family G proteins and recruits β-arrestins-1 and 2 (Fig. 1a). It was previously thought that the analgesic effects of μOR signalling were mediated by G-protein signalling[2], whereas respiratory depression was mediated by β-arrestin recruitment[3]. Thus, ligands that preferentially activate G protein, also known as G-protein-biased agonists, were expected to exhibit attenuated side effects. To this end, a series of G-protein-biased ligands were developed, including TRV130, PZM21, mitragynine pseudoindoxyl (MP) and SR-17018[4–8]. However, although ligand bias towards G-protein signalling leads to the reduction of β-arrestin-mediated tolerance, more recent studies have shown that overly strong G-protein signalling (super-efficacy) is responsible for respiratory depression[9–11], and that partial agonists with lower efficacy provide a safer therapeutic profile[12].

Some insight into the structural underpinnings of μOR activation and μOR-mediated G-protein signalling is provided by high-resolution structures. The C-terminal helix of G_i binds to an opening within the cytoplasmic surface of the 7-transmembrane helix bundle, which is formed upon an approximately 10-Å outward movement of the intracellular end of transmembrane helix 6[13–16] (TM6) (Fig. 1b). At present, there is still no high-resolution structure of μOR in complex with β-arrestin, probably owing to the lack of a stable or structurally homogenous protein complex. Nevertheless, structures determined by X-ray crystallography and cryo-electron microscopy (cryo-EM) generally represent snapshots of the most stable and homogenous conformations out of a large ensemble. The majority of GPCR–G-protein complex structures

have been determined in the nucleotide-free state, a highly stable state that may not represent the active state in the presence of the physiologic concentrations of GDP and GTP in cells[17]. Conformations of less stable excited states and their relative populations within the conformational ensemble may not be amenable to structure determination but represent important modulators of downstream signalling[18–21].

To investigate the molecular basis of μOR activation and signal transfer, we combined double electron–electron resonance (DEER) and single-molecule fluorescence resonance energy transfer (smFRET)[22–24]. DEER resolves an ensemble of conformations and their populations at sub-angstrom resolution and with high sensitivity to population changes, whereas smFRET provides access to real-time conformational dynamics. Here we examined the effect of nine representative μOR ligands with unique pharmacological profiles on the conformation and dynamics of TM6, including naloxone (antagonist), TRV130, PZM21, MP (low-efficacy G-protein-biased agonists), buprenorphine (low-efficacy agonist), morphine (high-efficacy agonist), DAMGO (high-efficacy reference agonist), BU72 and lofentanil (super-efficacy agonists) (Fig. 1c,e and Supplementary Fig. 1). Additionally, we investigated the synergistic effects of ligand and transducer binding on the conformational equilibrium and transducer activation–in particular nucleotide release from the G protein. Our results demonstrate how the conformational ensemble of μOR–whose conformational states exchange on fast and slow timescales–is fine-tuned by ligand binding, resulting in distinctive efficacies and signal bias.

## Nitroxide spin probe and fluorophore labelling

To label the μOR site-specifically with fluorophores or nitroxide spin labels, we first generated a minimal-cysteine μOR construct (μORΔ7),

[1]State Key Laboratory of Membrane Biology, Beijing Frontier Research Center for Biological Structure, Beijing Advanced Innovation Center for Structural Biology, Tsinghua University, Beijing, China. [2]Tsinghua–Peking Joint Center for Life Sciences, School of Medicine, Tsinghua University, Beijing, China. [3]School of Life Sciences, Tsinghua University, Beijing, China. [4]Jules Stein Eye Institute and Department of Chemistry and Biochemistry, University of California, Los Angeles, Los Angeles, CA, USA. [5]Institute for Drug Discovery, University of Leipzig Medical Center, Leipzig, Germany. [6]Department of Molecular and Cellular Physiology, Stanford University School of Medicine, Stanford, CA, USA. [7]Institute of Organic and Medicinal Chemistry, School of Pharmaceutical Sciences, University of Pécs, Pécs, Hungary. [8]Department of Anesthesiology, Washington University School of Medicine, St Louis, MO, USA. [9]These authors contributed equally: Jiawei Zhao, Matthias Elgeti. ✉e-mail: matthias.elgeti@uni-leipzig.de; kobilka@stanford.edu; chunlai@mail.tsinghua.edu.cn

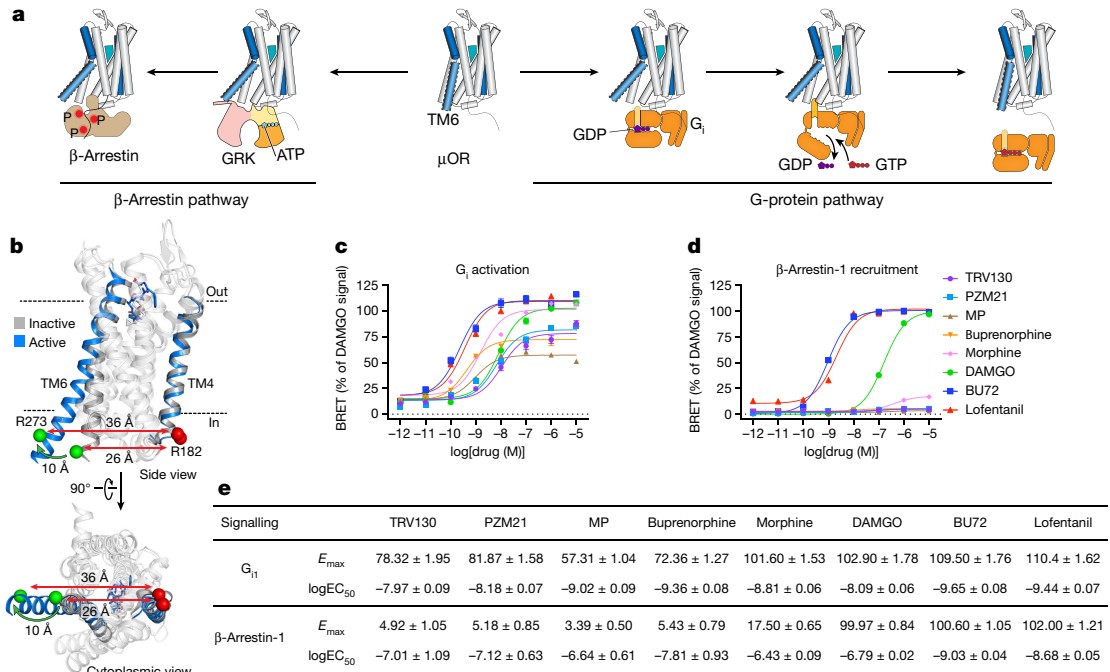

**Fig. 1 | Ligand-dependent activation of the μOR. a**, Binding of agonist to the μOR activates two downstream signalling pathways: the G-protein pathway and the β-arrestin pathway. **b**, The hallmark conformational change of GPCR activation is an outward tilt of TM6 of approximately 10 Å. Cα atoms of Arg182 in TM4 and Arg273 in TM6 are shown as red and green spheres, respectively. TM4 and TM6 are highlighted (inactive μOR (grey), Protein Data Bank (PDB)

4DKL; active, G-protein-bound μOR with G-protein hidden for clarity (blue), PDB code 6DDF). **c,d**, Intrinsic efficacy of ligands towards $G_{i1}$ and β-arrestin-1 determined by TRUPATH assays. Error bars represent s.e.m. from 9–12 biological replicates. **e**, Maximum efficacy ($E_{max}$) and potency (half-maximal effective concentration ($EC_{50}$)) values determined in **c,d**.

in which seven solvent-exposed cysteines were mutated to Ser, Thr, Ala or Leu, depending on the individual local environment (Extended Data Fig. 1). The μORΔ7 construct showed preserved function compared with the wild-type μOR in TRUPATH and ligand-binding assays (Extended Data Fig. 2). Furthermore, when reconstituted in lauryl maltose neopentyl glycol (LMNG) micelles, the purified μORΔ7 construct showed negligible background labelling of the remaining cysteines by the fluorophore (maleimide ATTO 488) or the nitroxide spin label HO-1427 (Extended Data Fig. 3). Two additional cysteine residues were introduced to the intracellular sides of TM4 and TM6 to create labelling sites for derivatization with spin-label or fluorophore reagents. The cysteine mutations did not significantly alter agonist or antagonist binding properties of the μOR (Extended Data Fig. 4). For DEER studies, μORΔ7(R182C/R276C) was derivatized with HO-1427 (creating μOR-HO-1427) (Extended Data Fig. 3i and Supplementary Fig. 2), a novel nitroxide spin label that combines the structures of two well-characterized spin labels, iodoacetamide proxyl and methanethiosulfonate spin label. HO-1427 generates a spin-label side chain characterized by reduced dynamics and a stable, non-reducible thioether bond[25]. For most smFRET studies, we labelled μORΔ7(R182C/R273C) and μORΔ7(T180C/R276C) with iodoacetamide-conjugated Cy3 and Cy5 fluorophore pair (Cy3/Cy5) and maleimide-conjugated Cy3 and Cy7 fluorophore pair (Cy3/Cy7), respectively, creating μOR–Cy3/Cy5 and μOR–Cy3/Cy7 (Extended Data Fig. 3b–g). Cy3/Cy5 and Cy3/Cy7 dye pairs exhibit different Förster radii (approximately 55 Å and 40 Å, respectively[26]), around which they are most sensitive to distance changes and the combination of both enables us to detect a large range of inter-dye distance changes with high sensitivity (Extended Data Fig. 3g).

## DEER reveals TM6 conformational heterogeneity

We examined TM4–TM6 distances of μOR by DEER under saturating ligand conditions and in the absence or presence of transducers

(nucleotide-depleted) $G_i$ or β-arrestin-1. Generic multi-Gaussian global fitting of the combined DEER data suggests a mixture of 6 Gaussians as the most parsimonious model describing the full datasets including all 30 conditions (Methods, Extended Data Fig. 5 and 6 and Supplementary Fig. 3). The resulting distance distributions and the populations (integrated areas) of the individual distance peaks are shown in Fig. 2. The two longest distances (45 Å and 57 Å) were excluded from the population analysis, since their populations were not correlated to the populations of other distance peaks (Extended Data Fig. 7) as expected for a ligand-dependent conformational equilibrium. These two distance peaks are likely to represent oligomeric or nonfunctional receptor populations.

Comparison with high-resolution structures suggests that the 33-Å peak represents a conformation with TM6 in an inactive, inward position, whereas the population of the 43-Å peak exhibits an outward tilted TM6, thus representing an active conformation (Extended Data Fig. 8). Correlation analysis revealed that populations around 26 Å and 33 Å, as well as those at 39 Å and 43 Å, are highly correlated ($P < 0.05$), dividing each, the inactive and active states, into two conformations (Extended Data Fig. 7). We refer to the inactive conformations centred around 26 Å and 33 Å as $R_1$ and $R_2$, and to the active conformations centred around 39 Å and 43 Å as $R_3$ and $R_4$. Previous DEER studies and molecular dynamics simulations of the β2-adrenergic receptor (β2AR) suggest that $R_1$ and $R_2$ represent inactive conformations with an intact and broken TM3–TM6 hydrogen bond, respectively[27–29].

## Modulation of conformational heterogeneity

According to its antagonistic properties in cellular assays, naloxone only weakly stabilized inactive $R_2$ at the cost of the active $R_3$ conformation (Fig. 2d). Instead, super-efficacy agonists BU72 and lofentanil quantitatively stabilized the active conformations $R_3$ and $R_4$ (Fig. 2a,d). Surprisingly, in the presence of low-efficacy G-protein-biased agonists

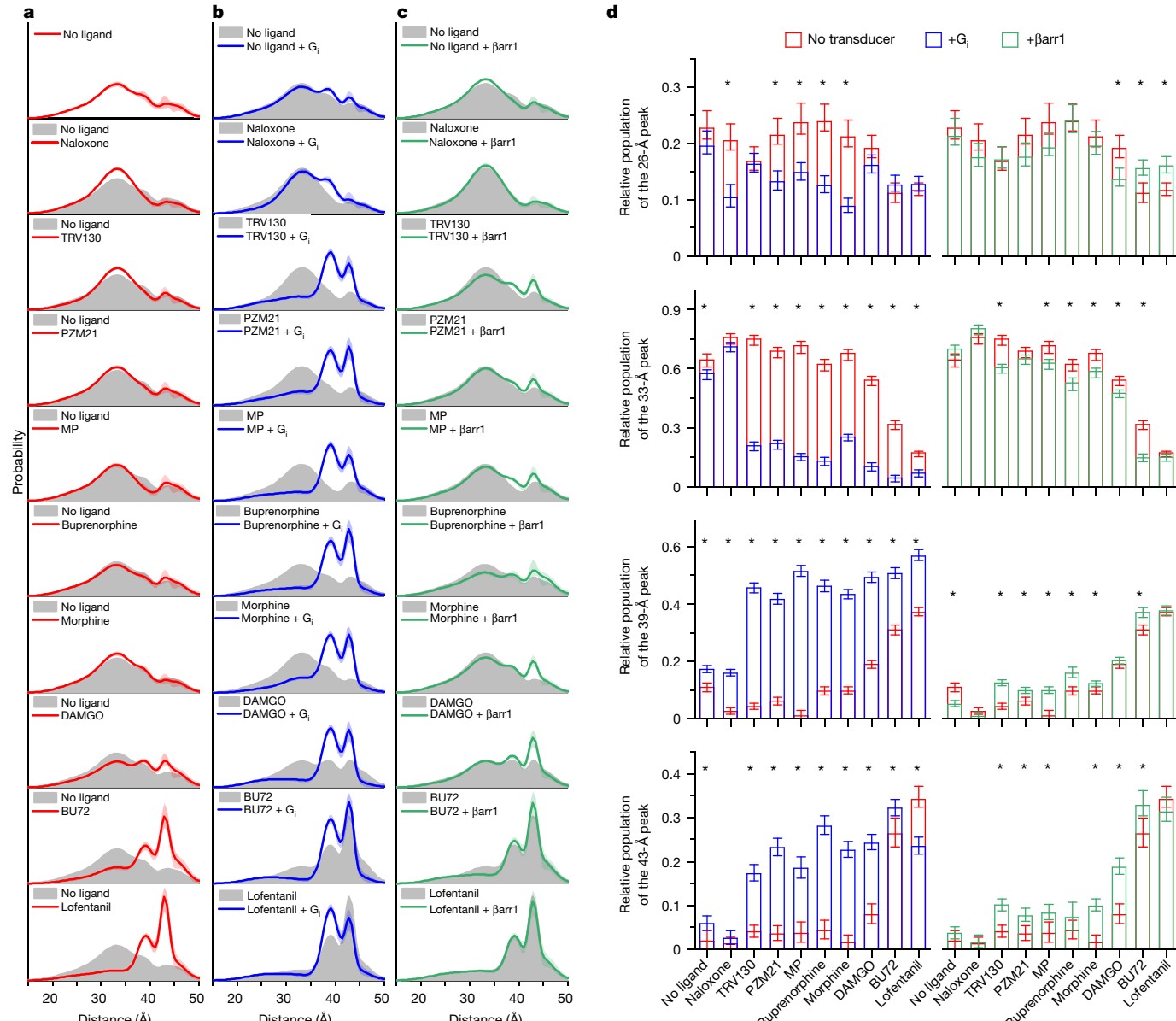

**Fig. 2 | Ligand- and transducer-dependent μOR conformational heterogeneity characterized by DEER. a**, Distance distributions of spin-labelled μOR under different ligand conditions. **b**, Distance distributions in the presence of ligand and $G_i$. **c**, Distance distributions of phosphorylated μOR (μORp) in the presence of ligand and pre-activated β-arrestin-1 (βarr1). **a**–**c**, Shaded areas along the line indicate 95% confidence interval. **d**, Gaussian populations centred around 26 Å, 33 Å, 39 Å and 43 Å. Data represent median population ± 95% confidence interval derived from bootstrapping analysis using $n = 1,000$ iterations. Populations marked with asterisks have non-overlapping confidence intervals in the presence and absence of transducer.

(TRV130, PZM21, MP and buprenorphine) the TM4–TM6 distance remained mostly in the inactive $R_1$ and $R_2$ conformations, suggesting that μOR regions other than TM6 control G-protein efficacy of these ligands (Fig. 1c). Binding of DAMGO, an analogue of the endogenous opioid met-enkephalin that is commonly used as the reference full agonist for the μOR, caused a small but significant population shift towards $R_3$ and $R_4$, in agreement with the higher efficacy of DAMGO compared with low-efficacy agonists. However, the discrepancy between the amount of the active conformations $R_3$ and $R_4$ (approximately 25%) and efficacy (100%) suggests that structural changes other than TM6 outward tilt are sufficient for permitting productive $G_i$ and β-arrestin-1 engagement.

Further evidence for $R_3$ and $R_4$ representing active conformations came from experiments in the presence of transducers, since $G_i$ as well as β-arrestin-1 bound and stabilized both conformations (Fig. 2b–d).

G-protein binding clearly revealed the class of G-protein-biased ligands (TRV130, PZM21 and MP) for which large fractions of active $R_3$ and $R_4$ were observed, with a slight preference for stabilizing $R_3$. For ligand-free and naloxone-bound μOR, the $G_i$-induced population shifts were much smaller. In the presence of the super-efficacious agonists BU72 and lofentanil, $R_3$ and $R_4$ were already dominant in the absence of a transducer, and the population shift from $R_4$ to $R_3$ confirmed preferential $G_i$ binding to $R_3$, at least under the chosen experimental conditions. The effect of β-arrestin-1 binding was much less pronounced: for non-biased agonists morphine and DAMGO, the most significant β-arrestin-1-induced population shifts were observed towards $R_4$ — however, β-arrestin-1 binding in the presence of G-protein-biased ligands was promiscuous towards $R_3$ and $R_4$ (Fig. 2c). In summary, the transducer-induced population shifts towards $R_3$ and $R_4$ reflect the

ability of bound ligand to stabilize specific transducer-binding conformations and thus their signalling bias towards G protein or β-arrestin-1.

## Ligand-specific conformational dynamics of μOR

To further investigate potential structural and functional differences between individual μOR conformations, we performed smFRET experiments. smFRET has been used to capture the conformational dynamics of β₂AR[30,31], metabotropic glutamate receptor dimer[32] and β-arrestin[33,34] in reconstituted systems or in cell membranes. We used an experimental design similar to that previously reported for β₂AR[30] and showed that smFRET of labelled μOR, despite the lower spatial resolution compared to DEER, provides access to protein dynamics and enables tight control of transducer and nucleotide conditions (Fig. 3a). Some ligand conditions had to be excluded from smFRET analysis: ligand-free μOR proved to be unstable under smFRET conditions, and the controlled substances buprenorphine, morphine and lofentanil were not available in China, where the smFRET experiments were performed.

All smFRET distributions recorded for Cy3 and Cy5-labelled μOR (μOR–Cy3/Cy5) could be described by one main Gaussian distribution (Fig. 3b) and a broad, ligand-independent distribution that probably represents noise. The position of the dominant fluorescence resonance energy transfer (FRET) peak was clearly ligand-dependent, which indicates that the time resolution (100 ms) was insufficient to resolve the transitions between at least two μOR conformations with distinct donor–acceptor distances. This resulted in time-averaged FRET efficiencies scaled by the populations of the underlying conformations (Supplementary Fig. 4a). The time-averaged FRET efficiencies were still able to distinguish the different ligands, as FRET efficiency progressively shifted from 0.89 to 0.77 in the presence of agonists of increasing efficacy, indicating an increase in the time-averaged fluorophore distance. Even though the difference in FRET peak centres between the antagonist naloxone and low-efficacy, G-protein-biased agonists TRV130, PZM21 and MP was small (Fig. 3b and Extended Data Fig. 9a), the average FRET values showed significant differences ($P < 0.001$; Extended Data Fig. 9b), indicating a small shift of the conformational equilibrium of μOR towards more open, active conformations in the presence of G-protein-biased agonists and full activation for DAMGO and BU72.

We also recorded smFRET data using the μOR–Cy3/Cy7 construct, whose fluorophore pair exhibits a shorter Förster radius than the Cy3 and Cy5 fluorophore pair (Extended Data Fig. 3g), and was attached to slightly altered labelling sites on μOR, using different labelling chemistry (Extended Data Fig. 10). Notably, for naloxone and the low-efficacy ligands TRV130, PZM21 and MP, the μOR–Cy3/Cy7 construct was able to resolve two well-separated FRET distributions, revealing a conformational exchange with an exchange rate slow enough to be captured by our smFRET setup (Fig. 3c). The high-FRET distribution was stably centred around 0.8 (blue), and dominant in the presence of antagonist naloxone and thus reflects an inactive conformation. The population of the low-FRET state (red) increased with G-protein efficacy of bound ligand, such that for the high-efficacy agonist DAMGO and the super-efficacy agonist BU72, only a low-FRET signal was observed. Further, the low-FRET distribution showed a ligand-dependent centre position below 0.7, indicating a time-averaged conformational equilibrium, similar to what we observed for μOR–Cy3/Cy5 (Fig. 3b, red).

We interpret these smFRET results as the superposition of two conformational changes: receptor-activating structural changes occurring on a fast timescale (<100 ms) lead to a ligand-dependent centre position of the associated FRET state observed with both constructs. This is in accordance with reports for other GPCRs, for which activation rates between 0.3–40 ms have been reported[28,35–37]. Additionally, and only observable using the μOR–Cy3/Cy7 construct, we identified a slow conformational transition (>100 ms). The underlying structural change

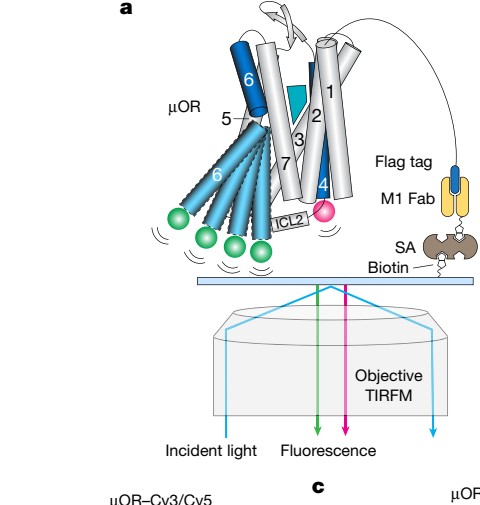

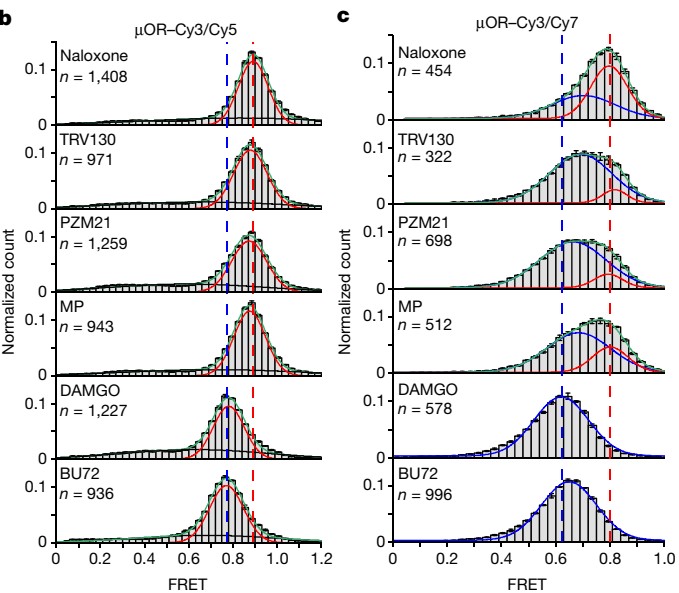

**Fig. 3 | SmFRET experiments of the μOR bound to different ligands.**
**a**, Schematic of single-molecule FRET experiment. Labelled μOR was tethered to a cover slip via its Flag tag, biotinylated M1 Fab, streptavidin (SA) and biotinylated PEG. TIRFM, total internal reflection fluorescence microscopy. **b,c**, SmFRET distributions of μOR–Cy3/Cy5 (**b**) and μOR–Cy3/Cy7 (**c**) in the presence of different ligands. Gaussian peaks were fitted to FRET states (red and blue) and background noise (black). Green lines represent the cumulative fitted distributions. Dashed lines in blue and red represent peak centres of naloxone- and DAMGO-bound samples, respectively (*n* represents the number of fluorescence traces used to calculate the corresponding histograms). Data are mean ± s.d. from three repeats.

reflects a prerequisite of G-protein binding or activation, as it clearly distinguishes μOR bound to naloxone from G-protein-biased ligands. We tentatively assign this slow conformational change to a structural transition in intracellular loop 2 (ICL2), which represents a critical receptor segment for G-protein binding and activation[38–40] and for which different conformations have been observed in high-resolution structures[41]. μOR–Cy3/Cy7 includes a labelling site at the C-terminal end of ICL2 (Extended Data Fig. 10c) and localized structural changes at equivalent site have been detected in a DEER study investigating ligand binding to the type 1 angiotensin II receptor[40] (AT1R). However, another possible interpretation for the slow conformational change includes a rotation of TM6, which represents a structural prerequisite of TM6 outward movement[42,43]. In any case, our smFRET findings complement our DEER results monitoring TM4–TM6 distances, in which DAMGO and G-protein-biased agonists had only a small or no significant

effect on the populations of active receptor species. Cy3/Cy5- and Cy3/Cy7-labelled μORΔ7(R182C/R276C), the same construct used in our DEER measurements (Extended Data Fig. 9c–h), displayed the similar trend of FRET changes in the presence of a series of ligands. However, μORΔ7(R182C/R276C)–Cy3/Cy7 is unable to resolve two FRET states shown in μORΔ7(R180/R276)–Cy3/Cy7 in the presence of low-efficacy ligands (Fig. 3c). This finding supports our assignment that these two FRET states reflect a slow conformational change of ICL2. Moving one labelling site from T180 to R182, thus away from ICL2, depletes the sensitivity towards local motions of ICL2. We attribute the discrepancy between smFRET and DEER to the long-linker fluorophores that may amplify the rotational conformation change and/or local conformational change to a linear distance change compared with the short spin labels (Extended Data Fig. 10).

## Conformational dynamics of μOR with G protein

To investigate the role of μOR conformational changes for transducer binding and nucleotide exchange, we examined μOR–Cy3/Cy5 in the presence of ligands and transducer. We chose μOR–Cy3/Cy5 over μOR–Cy3/Cy7 because of the higher signal-to-noise ratio of single-molecule fluorescence trajectories during these experiments to unambitiously characterize dynamic transitions between G-protein-bound and G-protein-unbound μOR. Compared with the active conformation stabilized by ligands alone (FRET efficiency of around 0.77; Fig. 3b), G-protein binding, upon depletion of nucleotide GDP using apyrase, led to a reduction in FRET efficiency to around 0.5 (Fig. 4a, blue and Supplementary Fig. 4b). This marked decrease may be owing to a direct interaction of G protein and fluorophore. The population of the low-FRET peak showed the same MP → TRV130 → PZM21 → DAMGO and BU72 progression as observed for ligand efficacy (Fig. 1) and is thus interpreted as nucleotide-free μOR–$G_i$ complex. The high-FRET peak (Fig. 4a, red) showed the same peak positions observed in the absence of G protein (Fig. 3b), and is thus interpreted as time-averaged equilibrium of active and inactive μOR conformations not bound to G protein. A third, ligand-independent and broad FRET distribution (Fig. 4a, black), is assumed to represent noise. Of note, the observation of two well-separated FRET peaks (centred around 0.5 and 0.8), representing G-protein-bound and G-protein-unbound μOR, respectively, provides the opportunity to apply a two-state hidden Markov Model[44] and to describe μOR complex formation and signal transfer in more detail. To this end, only traces that had at least one transition between high-FRET and low-FRET states during the course of the experiment were selected, thus enabling us to selectively analyse those μOR molecules involved in G-protein binding.

To characterize conformational dynamics of GDP-bound and nucleotide-free forms of μOR–$G_i$ complex, we recorded smFRET time traces at different concentrations of GDP (Fig. 4b and Supplementary Fig. 5). We found that for high- and super-efficacy agonists DAMGO and BU72 the low-FRET peak population was reduced with increasing GDP concentrations (Fig. 4c and Extended Data Fig. 11), indicating dissociation of the μOR–$G_i$•GDP complex and reestablishment of the time-averaged, ligand-bound μOR state (Fig. 3b). For these two ligands, we also observed a shift of the low-FRET peak from around 0.5 to 0.6 with increasing GDP concentration (Fig. 4d), and we assign the 0.6 low-FRET state to the complex of active μOR with GDP-bound $G_i$ as opposed to the nucleotide-free complex at around 0.5 (Fig. 4e). Similar smFRET changes were described to occur transiently for GDP-bound $G_s$ interacting with β2AR[30]. In contrast to the high-efficacy and super-efficacy agonists, the 0.6 FRET state was dominant for low-efficacy G-protein-biased agonists at all GDP concentrations, indicating increased stability of the GDP-bound μOR–$G_i$ complex for these ligands (Fig. 4d and Extended Data Fig. 11).

On the basis of previous studies[30,45], we used a simplified, three-state model of G-protein binding to active μOR (Fig. 4e) for the evaluation of the dwell-time distributions of high- and low-FRET states (Supplementary Figs. 6 and 7). The dwell-time distributions of the high-FRET state were adequately described by mono-exponentials indicating a single rate-limiting step of G-protein binding (Supplementary Fig. 6). The resulting high-FRET dwell times are shown in Fig. 4f and indicate that the rate of G-protein binding is largely independent of GDP for all ligands. However, for DAMGO and BU72, both of which quantitatively stabilized the μOR–$G_i$ complex in the absence of nucleotide (Fig. 4a), overall shorter high-FRET dwell times indicate faster binding of $G_i$ to μOR compared with low-efficacy G-protein-biased agonists (Fig. 4f). The rates of G-protein binding scaled with the amount of active μOR, as identified by smFRET in the absence of G protein (Fig. 3b,c).

The dwell-time distributions of the low-FRET state are associated with two low-FRET states at 0.6 and 0.5 FRET, reflecting the GDP-bound and nucleotide-free μOR–$G_i$ complex, respectively (Fig. 4e). Correspondingly, for all ligands, the low-FRET dwell-time distributions were best described using biexponential decay curves (Supplementary Fig. 7), and for simplicity, we calculated a weighted average of low-FRET dwell times for each condition to represent the overall stability of the μOR–$G_i$ complex (Fig. 4g). At a physiological GDP concentration of 30 μM, low-FRET dwell times for all ligands were very similar. At low GDP concentration and only in the presence of high-efficacy agonists DAMGO and BU72, longer low-FRET dwell times indicated a higher stability of the nucleotide-free μOR–$G_i$ complex. Together, these results show that G-protein-biased agonists do not lower GDP affinity to $G_i$ as much as high-efficacy and super-efficacy agonists, which, in combination with slower $G_i$ binding (Fig. 4f), manifests in their lower efficacy.

Similar to the results of our DEER experiments, which showed only subtle population shifts due to β-arrestin-1 binding to μORp (Fig. 2c), the smFRET distributions of μORp–Cy3/Cy5 show very little effect in response to β-arrestin-1 binding (Extended Data Fig. 12). These data support the current understanding of a promiscuous, low-affinity interaction of the arrestin finger loop with active GPCR conformations and suggests the necessity of this 'core engagement' for stabilization of an active, low-FRET conformation[46].

## Conclusion

The present study reveals differences in the structure and dynamics of μOR bound to functionally diverse ligands and the effects of these differences on receptor catalytic activity and stability of the receptor–transducer complex. Our findings characterize the molecular underpinnings of $G_i$ activation and β-arrestin-1 recruitment and provide insight into the mechanism of super-efficacy agonism, which cannot be understood on the basis of static X-ray and cryo-electron microscopy structures alone. Previous studies using NMR spectroscopy, molecular dynamics simulations, and DEER indicate that the conformational dynamics of GPCRs, especially in the TM5, TM6, TM7, ICL1, ICL2 and H8 domains[40,47–49], have important roles in functional selectivity of GPCRs. Our results reveal the conformational heterogeneity of TM6 and that both fast and slow conformational dynamics of TM6 and ICL2 are differentially modulated by distinct ligands.

We performed DEER experiments, which highlight the conformational heterogeneity of μOR and how the ensemble of conformations is modulated by ligands with distinct functions. For low-efficacy G-protein-biased agonists we did not observe significant populations of receptor in the canonical 'active' conformation, which includes the outward tilt of TM6. However, the addition of the transducers $G_i$ and β-arrestin-1 clearly revealed that these ligands 'pre-activate' the receptor, thereby facilitating transducer binding. Additionally, DEER was able to resolve two active conformations of TM6, for which our results suggest distinct G-protein affinities. In accordance with existing studies, binding of β-arrestin-1 to the intrahelical transducer binding site of μORp (core interaction) is more promiscuous and occurs with lower affinity.

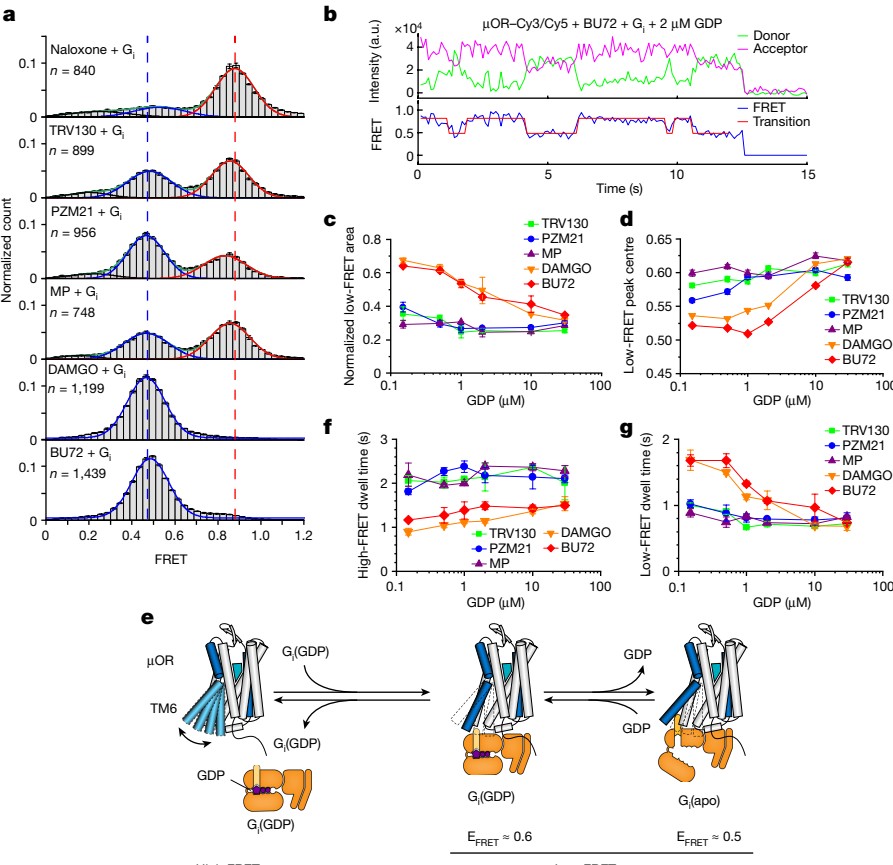

**Fig. 4 | Structural dynamics of the μOR in the presence of $G_i$ and GDP.**
**a**, smFRET distributions of μOR–Cy3/Cy5 in the presence of different ligands and $G_i$, followed by treatment of apyrase to remove GDP. Red, blue and black lines represent Gaussians fitted to high-FRET, low-FRET and nonfunctional states, respectively. Green lines represent the cumulative fitted distributions. Dashed lines indicate high-FRET peak centre of naloxone sample (red) and low-FRET peak centre of the BU72 sample (blue), respectively. *n* represents the number of fluorescence traces used to calculate the corresponding histograms. Data are mean ± s.e.m. from three repeats. **b**, Exemplary smFRET traces of μOR–Cy3/Cy5 and analysis via a two-state hidden Markov model. a.u., arbitrary units. **c**, Area of the low-FRET peak at increasing GDP concentrations. Data are mean ± s.d. from two biological repeats. **d**, Low-FRET peak position with increasing GDP concentrations. Frames of low-FRET state identified by a two-state hidden Markov Model were extracted and binned to plot histograms. FRET histograms were further fitted to Gaussians and the peak centres are plotted. Error bars represent the standard error of fitting. **e**, Schematic of a simplified reaction model of G-protein coupling. **f**, Dwell time of the high-FRET state. **g**, Dwell time of the low-FRET state. **f**,**g**, Data are mean ± s.d. from two biological repeats.

The discrepancy between the canonical active receptor population observed in DEER and ligand efficacy, which is especially apparent for DAMGO, suggests that TM6 movement alone does not define receptor activity. We used smFRET as a complementary method as it provides access to rates of conformational interconversion, which have been implicated as 'kinetic controls' of G-protein binding or activation in other GPCRs[37,50]. The specific properties of the chosen fluorophores and receptor-labelling sites prove vital for capturing activating conformational changes at the intracellular receptor surface that correlate with the efficacy of bound ligand. Our data revealed a slow conformational change with an exchange dwell time of more than 100 ms connected to receptor pre-activation, a structural change that distinguishes μOR bound to the antagonist naloxone and low-efficacy G-protein-biased agonists, which is a potentially rate-limiting step for G-protein and β-arrestin binding and signalling. Experiments conducted in the presence of G protein and various concentrations of nucleotide GDP enabled the identification of the GDP-bound and nucleotide-free ternary complexes and how their formation is modulated by the nature of bound ligand. Even though 'pre-activated' μOR may bind G protein efficiently enough to cause signalling, fully activated μOR, as present in high- and super-efficacy bound μOR, couples to $G_i$ at twice the rate. Moreover, once the ternary complex is formed, high-efficacy and super-efficacy agonists lower the affinity towards GDP substantially, thereby driving GDP release and G-protein activation. Low-efficacy, G-protein-biased agonists lead to a slower release of GDP and large fractions of the complex remain GDP-bound. Thus, the rate of G-protein binding and GDP release are both ligand-controlled via modulation of the conformational ensemble involving inactive, pre-activated and fully activated species. Instead, binding of β-arrestin-1 to the receptor core relies on formation of the canonical, fully activated receptor conformation as binding of low-efficacy, G-protein-biased agonists promotes formation of the μOR–β-arrestin-1 complex only weakly, whereas we observed greater changes for the more efficacious morphine and DAMGO. Of interest, when bound to lofentanil and BU72, μOR exists mostly in the active conformations, in agreement with their high efficacy for recruitment of β-arrestin-1; however, since no significant change in the DEER distributions was observed upon the addition of β-arrestin-1, we cannot conclude that it actually bound.

In sum, this study provides insights into μOR functional selectivity and super-efficacy, based on the coexistence and differential population of inactive and active conformations exchanging on fast or slow timescales. Moreover, it emphasizes the importance of solution-state, biophysical studies for the characterization of GPCR–ligand–transducer signalling, as we report experimental evidence for important

intermediate conformations that are responsible for G-protein functional selectivity. These findings suggest potential approaches for the design of therapeutic agents with fewer adverse effects, that target sparsely populated conformational states that have evaded detection by high-resolution structural biology methods. The need for such therapies is imminent for the opioid receptor subfamily, but intermediate conformations with functional selectivity properties have been reported for other GPCRs[40], and thus this approach may be generalizable for other targets.

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

# Methods

## μOR expression and purification

The wild-type *Mus musculus* μOR (6-398) with an N-terminal HA signal sequence followed by a Flag tag and a C-terminal 8×His tag was cloned in the pFastBac1 vector. The minimal-cysteine construct (μORΔ7) was created by introducing the mutations[51] C13S, C22S, C43S, C57S, C170T, C346A and C351L into the wild-type μOR. Double-cysteine mutation constructs (μORΔ7(R182C/R276C) for DEER, μORΔ7(T180C/R276C) and μORΔ7(R182C/R273C) for smFRET experiments) were generated based on the μORΔ7 construct. The μOR was expressed and purified following a previous protocol[13] with some modifications. The μOR was expressed in Sf9 insect cells (Expression Systems, authenticated by supplier, not tested for mycoplasma) using Bac-to-Bac baculovirus systems with 10 μM naloxone. Cells were collected 48 h post infection and were lysed in a buffer of 10 mM Tris pH 7.5, 1 mM EDTA, 100 μM TCEP, 10 μM naloxone, 160 μg ml$^{-1}$ benzamidine and 2.5 μg ml$^{-1}$ leupeptin. The receptor was extracted from the Sf9 membrane using buffer of 20 mM HEPES pH 7.5, 500 mM NaCl, 0.7% *N*-dodecyl-β-D-maltoside (DDM), 0.3% CHAPS, 0.03% cholesteryl hemisuccinate (CHS), 30% (v/v) glycerol, 5 mM imidazole, 2 mM MgCl$_2$, 160 μg ml$^{-1}$ benzamidine, 2.5 μg ml$^{-1}$ leupeptin, 10 μM naloxone, 100 μM TCEP and 2 μl benzonase in the cold room for 1 h. After centrifugation, Ni-NTA resin was added to the supernatant in a 500-ml centrifuge tube (Corning) and rotated for 2 h at 4 °C. Ni-NTA resin was washed in batch with washing buffer of 20 mM HEPES pH 7.5, 500 mM NaCl, 0.1% DDM, 0.03% CHAPS, 0.03% CHS, 5 mM imidazole and 10 μM naloxone and protein was eluted in washing buffer supplemented with 250 mM imidazole. Ni-NTA eluate was supplemented with 2 mM CaCl$_2$ and loaded onto anti-Flag M1 resin (Millipore-Sigma) for further purification. The detergent was exchanged to LMNG on a Flag column by gradually increasing the proportion of the exchange buffer (20 mM HEPES pH 7.5, 100 mM NaCl, 0.5 LMNG, 0.05% CHS, 2 mM CaCl$_2$ and 10 μM naloxone) over the Ni-NTA washing buffer supplemented with 2 mM CaCl$_2$ at room temperature for 1 h. The μOR was finally eluted with buffer of 20 mM HEPES pH 7.5, 100 mM NaCl, 0.01% LMNG, 0.001% CHS, 5 mM EDTA, 0.2 mg ml$^{-1}$ Flag peptide and 10 μM naloxone. After concentrating with a 4-ml 100-kDa cutoff concentrator (Amicon Ultra), the μOR was further purified by size-exclusion chromatography (SEC) using an SD200 increase 10/300 column (GE Healthcare) equilibrated with SEC buffer of 20 mM HEPES pH 7.5, 100 mM NaCl, 0.01% LMNG, 0.001% CHS and 10 μM naloxone. Fractions containing monomeric μOR were collected and concentrated with a 500-μl 100-kDa cutoff concentrator (Amicon Ultra). The μOR was supplemented with 15% (v/v) glycerol and flash frozen in liquid nitrogen.

## G$_i$ heterotrimer expression and purification

DNA for the human Gα$_{i1}$ was cloned into the pFastBac1 vector. DNA of human Gβ$_1$ with an N-terminal 6×His tag and HRV 3 C protease cleavage site (LEVLFQGP) and Gγ$_2$ were cloned into the vector of pFastBac Dual under the promoter of ph and p10, respectively. P2 viruses of Gα$_{i1}$ and Gβ$_1$γ$_2$ were generated following the same protocol for the μOR. G$_{i1}$ heterotrimer was expressed in Hi5 cells (Expression Systems, authenticated by supplier, not tested for mycoplasma) with 4 ml P2 of Gα$_{i1}$ and 10 ml P2 of Gβ$_1$γ$_2$ per liter cells when cells reached a density of 3 million per ml. Cells were collected 48 h post infection and kept in −80 °C freezer until use.

Cell pellets were thawed in lysis buffer (10 mM Tris pH 7.5, 1 mM MgCl$_2$, 5 mM β-mercaptoethanol (β-ME), 10 μM GDP, 160 μg ml$^{-1}$ benzamidine, 2.5 μg ml$^{-1}$ leupeptin). After centrifugation, pellets were solubilized in solubilization buffer (20 mM HEPES pH 7.5, 100 mM NaCl, 1% sodium cholate, 0.05% LMNG, 5 mM MgCl$_2$, 20 mM imidazole, 5 mM β-ME, 10 μM GDP, 160 μg ml$^{-1}$ benzamidine, 2.5 μg ml$^{-1}$ leupeptin) and were stirred in a cold room for 1 h. After centrifugation at 14,000 rpm for 20 min, the supernatant was mixed with Ni-NTA resin and rotated at 4 °C for 1 h. Ni-NTA resin was then washed four times in batch with solubilization buffer. Detergent was exchanged to LMNG on the Ni-NTA column by gradually increasing LMNG concentration at room temperature. Protein was eluted with elution buffer (20 mM HEPES pH 7.5, 50 mM NaCl, 0.01% LMNG, 2 mM MgCl$_2$, 5 mM β-ME, 10 μM GDP, 180 mM imidazole). The His tag was cleaved by 1:50 (w/w) HRC 3 C protease. G$_{i1}$ was treated with 5 μl of λ protein phosphatase and was dialysed against dialysis buffer (20 mM HEPES pH 7.5, 50 mM NaCl, 0.01% LMNG, 2 mM MgCl$_2$, 2 mM MnCl$_2$, 5 mM β-ME, 10 μM GDP) overnight at 4 °C to remove imidazole. The His tag and contaminates were removed by loading G$_{i1}$ onto 2-ml Ni-NTA resin. Flow-through of Ni-NTA resin was loaded onto a MonoQ column and G$_{i1}$ was further purified by anion exchange. The G$_{i1}$ heterotrimer peak was collected and concentrated. After being supplemented with 15% glycerol, G$_{i1}$ was flash froze and kept in −80 °C freezer. For DEER samples, ion-exchange purified G$_{i1}$ was further injected onto an SD200 increase 10/300 column (GE Healthcare) equilibrated with SEC buffer (20 mM HEPES pH 7.5, 100 mM NaCl, 0.01% LMNG, 2 mM MgCl$_2$ and 10 μM GDP). SEC fractions were pooled, concentrated to 336 μM and flash frozen.

## GRK5 expression and purification

Human GRK5 DNA with a C-terminal 6×His tag was cloned into pFast-Bac1 vector. P2 virus was generated following the same protocol of the μOR. GRK5 was expressed in Sf9 insect cells with 25 ml of P2 virus and was collected 48 h after infection. Purification of GRK5 was performed on ice or at 4 °C. Cells were lysed in lysis buffer (20 mM HEPES pH 7.5, 150 mM NaCl, 20 mM imidazole, 5 mM β-ME, 160 μg ml$^{-1}$ benzamidine, 2.5 μg ml$^{-1}$ leupeptin) by sonication on ice. Cell debris was removed by centrifuge at 14,000 rpm for 20 min. GRK5 in supernatant was purified by Ni-NTA resin using wash buffer (20 mM HEPES pH 7.5, 150 mM NaCl, 20 mM imidazole, 5 mM β-ME). Protein was eluted in wash buffer supplemented with 160 mM imidazole. GRK5 was concentrated and injected in an SD200 increase 10/300 column equilibrated with cold SEC buffer (20 mM HEPES pH 7.5, 300 mM NaCl) in cold room. SEC fractions of GRK5 were pooled, concentrated and flash frozen.

## β-Arrestin-1 expression and purification

To investigate the conformational changes of the μOR in the presence of β-arrestin-1, a C-terminal truncated β-arrestin-1 was used for smFRET and DEER measurements. The long splice variant of human, cysteine-free (C59V/C125S/C140L/C150V/C242V/C251V/C269S), truncated β-arrestin-1 (1-382) (βarr1(ΔCT))[52] with an N-terminal 6×His and HRV 3 C site was in vector of pET15b and was transformed into BL21 (DE3) competent cells. *Escherichia coli* cells were cultured in TB medium with 100 μg ml$^{-1}$ ampicillin until OD$_{600}$ reached 1.2 at 37 °C in a shaker at 220 rpm. The temperature was decreased to 18 °C and protein expression was induced with 200 μM IPTG for 16 h. Purification of βarr1(ΔCT) was performed on ice or at 4 °C. Cells were collected and sonicated in buffer 1 (20 mM Tris 8.0 (25 °C), 300 mM NaCl, 20 mM imidazole) supplemented with 160 μg ml$^{-1}$ benzamidine and 2.5 μg ml$^{-1}$ leupeptin. After centrifugation, protein in the supernatant was incubated with Ni-NTA resin at 4 °C for 1 h. The Ni-NTA resin was extensively washed with buffer 1, then was further washed with 3 column volumes of buffer 2 (20 mM Tris 8.0 (25 °C), 50 mM NaCl and 20 mM imidazole). βarr1(ΔCT) was eluted with buffer 2 supplemented with 160 mM imidazole. βarr1(ΔCT) was loaded onto a Source 15Q 4.6/100 PE anion-exchange column (GE Healthcare). The column was washed with 2 column volumes of buffer A (20 mM Tris 8.0 (25 °C), 50 mM NaCl), and βarr1(ΔCT) was eluted with 15 column volumes of a linear gradient from 0 to 30% buffer B (20 mM Tris 8.0 (25 °C), 1 M NaCl). The peak fractions were pooled and supplemented with NaCl to a final concentration of 300 mM, which prevented the protein from precipitating when concentrated to high concentration in the following step. The protein was concentrated and injected in an SD200 increase 10/300 column equilibrated with SEC buffer of 20 mM HEPES pH 7.5, 300 mM NaCl. For DEER samples, SEC buffer was made in D$_2$O, and βarr1(ΔCT) was concentrated to 986 μM and flash frozen.

## Phosphorylation of μOR

The μOR was purified following the standard μOR purification protocol except that the naloxone was replaced with 10 μM DAMGO on the anti-Flag M1 resin and SEC purification procedures. 4 μM of μORΔ7(R182C/R276C) purified in the presence of DAMGO was incubated in phosphorylation buffer of 20 mM HEPES pH 7.5, 35 mM NaCl, 5 mM MgCl₂, 100 μM TCEP, 20 μM 1,2-dioctanoyl-*sn*-glycero-3-phospho-(1′-myo-inositol-4′,5′-bisphosphate) (C8-PIP2), 0.01% LMNG, 0.001% CHS and 100 μM DAMGO at room temperature for 1 h. ATP and GRK5 were then added to the reaction to a final concentration of 1 mM and 0.8 μM, respectively, and incubated for 1 h before more GRK5 was added. GRK5 was added every 1 h four times in total and the reaction was kept at room temperature.

To evaluate the phosphorylation level and make sure it reaches completion using ion-exchange chromatography, 12 μl of the phosphorylation reaction containing about 50 picomoles of μOR at different time points was removed and diluted to 200 μl using the buffer of 20 mM Tris pH 8.0 (25 °C), 50 mM NaCl, 0.01% LMNG, 5 mM EDTA and 10 μM naloxone. The samples were then injected onto a MonoQ (5/50) anion-exchange column (GE Healthcare) equilibrated with buffer A of 20 mM Tris 8.0 (25 °C), 50 mM NaCl, 0.01% LMNG and 10 μM naloxone. The column was washed with 1 column volumes of buffer A, and then with 40 column volumes of a linear gradient from 0 to 40% buffer B of 20 mM Tris 8.0 (25 °C), 1 M NaCl, 0.01% LMNG and 10 μM naloxone at room temperature. Protein elution was monitored by a fluorescence detector (Shimadzu) with excitation at 280 nm and emission at 340 nm (Extended Data Fig. 12a).

After the 4-h incubation with GRK5, the reaction was diluted by tenfold with the wash buffer of 20 mM HEPES pH 7.5, 100 mM NaCl, 0.01% LMNG, 0.001% CHS, 2 mM CaCl₂ and 10 μM naloxone before loading onto 3 ml M1 resin. The M1 resin was washed with 30 ml of the wash buffer at room temperature for 30 min. The μOR was finally eluted using elution buffer of 20 mM HEPES pH 7.5, 100 mM NaCl, 10 μM naloxone, 5 mM EDTA and 0.2 mg ml⁻¹ Flag peptide. After concentration, the μOR was further injected onto an SD200 increase 10/300 column equilibrated with SEC buffer of 20 mM HEPES pH 7.5, 100 mM NaCl, 0.01% LMNG, 0.001% CHS and 10 μM naloxone. Fractions containing monomeric μOR were collected and concentrated with a 500-μl 100-kDa cutoff concentrator (Amicon Ultra). The μOR was supplemented with 15% (v/v) glycerol and flash frozen in liquid nitrogen.

## Fluorophore synthesis

Iodoacetamide-conjugated Cy3 and Cy5 fluorophores were synthesized following a previous protocol[30]. In brief, 1 μmol of sulfo-Cyanine3 NHS ester or sulfo-Cyanine5 NHS ester (Lumiprobe) was dissolved in 500 μl dry dimethyl sulfoxide (DMSO). It was then added dropwise to a solution of 50 μl cadaverine in 500 μl of dry DMSO at room temperature. The reaction solution was stirred at room temperature for 5 min, then poured into 15 ml of 5% formic acid in ethyl acetate. The precipitate was collected and purified by high-performance liquid chromatography using 10 mM triethylammonium acetate pH 7.0 aqueous buffer (solvent A) with 100% acetonitrile (solvent B) as the mobile phase. The product fraction was dried using a rotary evaporator. The resulting pure fluorophore–cadaverine compound was then dissolved in 1 ml dry DMSO. *N*,*N*-diisopropylethylamine (100 μl) was added to this solution, followed by 1 mg iodoacetic acid NHS ester. The reaction solution was stirred at room temperature for 15 min and then poured into 15 ml ethyl acetate. The precipitate was collected and purified by high-performance liquid chromatography.

## Synthesis of HO-1427

The bromo derivative[53] (261 mg, 1.0 mmol) (HO-559) was dissolved in acetone (20 ml) and NaI (300 mg, 2 mmol) was added. The reaction mixture was refluxed for 1 h then evaporated. The residue was dissolved

in ethyl acetate/diethyl ether (50:50, 20 ml) and washed with brine (2 × 10 ml). The organic phase was dried (MgSO₄), filtered, evaporated and purified with flash chormatography (hexane:diethyl ether) yielding yellow crystals 230 mg (74%); melting point: 132–134 °C; retention factor ($R_f$) = 0.4 (hexane:ethyl acetate 2:1); Elemental analysis calculated for $C_{10}H_{15}INO_2$ (Mw: 308.1) C: 38.98; H: 4.91; N: 4.55%; measured: C: 39.02; H: 4.78; N: 4.61%; IR (cm⁻¹): 1665, 1615; MS (EI, m/z,%): 308 (8), 294 (6), 278 (6), 151 (100), 136 (8), 109 (52), 43 (61).

The melting point was measured with a Boetius micro melting point apparatus. The infrared (IR) spectrum was obtained using a Bruker Alpha FT-IR instrument with an attenuated total reflectance support on a diamond plate. The mass spectrum was recorded on a Shimadzu GCMS-2020 spectrometer in electron ionization (EI) mode (70 eV). The elemental analysis was performed on a Fisons EA 1110 CHNS instrument. Flash column chromatography was performed on Merck Kieselgel 60 (0.040–0.063 mm) column. Qualitative thin layer chromatography (TLC) was carried out on commercially available plates (20 cm × 20 cm × 0.02 cm) coated with Merck Kieselgel.

## μOR labelling with fluorophores

Minimal-cysteine μOR with cysteine mutations on TM4 and TM6, namely μORΔ7(T180C/R276C) and μORΔ7(R182C/R273C), was labelled by commercial maleimide-conjugated sulfo-Cy3 and sulfo-Cy7 (Lumiprobe) or by home-made iodoacetamide-conjugated Cy3 and Cy5, respectively. SEC purified μOR was diluted to 10 μM in 20 μl of labelling buffer (50 mM HEPES pH 7.5, 100 mM NaCl, 0.01% LMNG, 0.001% CHS, 10 μM naloxone). 30 μM of donor fluorophore and 60 μM of acceptor fluorophore were added into the reaction. After incubation at 20 °C for 30 min, free dyes were quenched with 10 mM L-cysteine. The reaction was then loaded onto a home-packed desalt column filled with 2-ml G50 resin (Sigma) equilibrated with the desalt buffer (20 mM HEPES pH 7.5, 100 mM NaCl, 0.01% LMNG, 0.001% CHS, 15% glycerol). Fractions containing μOR were pooled, aliquoted and flash frozen. The concentration of μOR was approximately 500 nM.

## μOR labelling with nitroxide spin label

To make samples of the μOR alone or in complex with G protein for DEER studies, SEC purified μORΔ7(R182C/R276C) without phosphorylation was diluted to 20 μM in labelling buffer (20 mM HEPES pH 7.5, 100 mM NaCl, 0.01% LMNG, 0.001% CHS, 10 μM naloxone). Nitroxide spin label reagent HO-1427 was added to a final concentration of 400 μM. After incubation at room temperature for 3 h, the reaction was quenched with 5 mM L-cysteine and was injected into an SD200 increase 10/300 column equilibrated with SEC buffer (20 mM HEPES pH 7.5, 100 mM NaCl, 0.01% LMNG, 0.001% CHS, 2 mM CaCl₂ in D₂O). Fractions of the monodisperse peak were pooled and equally divided into ten 1.5-ml tubes. The protein was diluted fourfold with SEC buffer. Ligands were added to each tube at a final concentration of 1 mM for naloxone, TRV130, PZM21, MP, buprenorphine, and morphine, 400 μM for DAMGO, 200 μM for lofentanil, and 500 μM for BU72. One tube of protein was kept without ligand. The μOR and ligand were incubated at room temperature for 2 h. Protein in each individual tube was concentrated and split into two parts, one of which was mixed with 20% (v/v) D8-glycerol, transferred to a capillary, and flash frozen. The other part was mixed with a threefold molar excess of G_{i1}, which was purified in D₂O buffer, and incubated for 30 min at room temperature. 1:100 apyrase (v/v, NEB) was added to the G-protein samples to remove free GDP and incubated for 1 h at room temperature. The G-protein samples were then mixed with 20% (v/v) D8-glycerol, transferred to capillaries and flash frozen.

To make samples in complex with βarr1(ΔCT) for DEER studies, μORpΔ7(R182C/R276C) was labelled with HO-1427 following a similar protocol above. SEC fractions were pooled and equally divided into 10× 1.5-ml tubes. The protein was diluted fourfold with D₂O dilution buffer of 20 mM HEPES pH 7.5, 100 mM NaCl, 0.01% LMNG, 0.001% CHS, 5 μM C8-PIP2, and respective ligand at a final concentration as

indicated above. The μOR was incubated with ligand for 2 h at room temperature. Protein was then concentrated, mixed with a fourfold molar excess of βarr1(ΔCT) that was in $D_2O$ buffer, and incubated at room temperature for 1 h. The samples were then mixed with 20% (v/v) D8-glycerol, transferred to capillaries and flash frozen.

### Single-molecule FRET experiments and analysis

All smFRET experiments were performed at 25 °C following previous protocol with some modifications[54]. In brief, single-molecule FRET studies were performed on a home-built objective-type TIRFM microscope, based on a Nikon Eclipse Ti-E with an EMCCD camera (Andor iXon Ultra 897), and solid-state 532 nm excitation lasers (Coherent Inc. OBIS Smart Lasers). Fluorescence emission from the probes was collected by the microscope and spectrally separated by interference dichroic (T635lpxr, Chroma) and bandpass filters, ET585/65 m (Chroma, Cy3) and ET700/75 m (Chroma, Cy5), in a Dual-View spectral splitter (Photometrics). No bandpass filter was used for Cy7 in the Dual-View spectral splitter. The hardware was controlled and smFRET movies were collected using Cell Vision software (Beijing Coolight Technology).

The μOR was immobilized on the cover slip via biotinylated M1 Fab and streptavidin. In brief, the assembled glass chamber, which had been cleaned and passivated with biotin-polyethylene glycol, was incubated with 0.05 mg ml$^{-1}$ streptavidin in 20 mM HEPES 7.5, 100 mM NaCl. One minute later, the unbound streptavidin was washed out by 25 nM biotinylated M1 Fab in incubation buffer (50 mM HEPES pH 7.5, 100 mM NaCl, 0.01% LMNG, 0.001% CHS, 2 mM CaCl$_2$, 5 mM MgCl$_2$ and 100 μM ligand). The biotinylated M1 Fab was incubated in the channel for one minute and the unbound M1 Fab was washed out by incubation buffer. The N-terminal Flag-tagged, fluorophore-labelled μOR was diluted to around 20 nM in incubation buffer and incubated on ice for 1 h before measurement. The μOR was diluted to about 1 nM and injected into the chamber. The unbound μOR was removed by imaging buffer (incubation buffer + 50 nM protocatechuate-3,4-dioxygenase (PCD), 2.5 mM protocatechuic acid (PCA), 1.5 mM aged Trolox, 1 mM 4-nitrobenzyl alcohol (NBA), 1 mM cyclooctatetraene (COT)). Movies were taken at a frame rate of 10 s$^{-1}$ using the Cell Vision software. For measurement in complex with GDP-free G$_{i1}$, 20 nM μOR in the presence of 100 μM ligand was incubated with 20 μM G$_{i1}$ for 30 min followed by addition of 1:100 (v/v, NEB) apyrase. After incubation on ice for 1 h, the complex was diluted and injected into the chamber and measured following the same protocol above. For measurement in the presence of G$_{i1}$ and GDP, the surface-immobilized μOR was incubated with imaging buffer, then 20 μM G$_{i1}$ and various concentrations of GDP in imaging buffer were injected into the chamber and imaged. For measurement in the presence of βarr1(ΔCT), the phosphorylated, Cy3/Cy5-labelled μOR was diluted to about 20 nM in arrestin buffer (50 mM HEPES pH 7.5, 100 mM NaCl, 0.01% LMNG, 0.001% CHS, 2 mM CaCl$_2$, 5 mM MgCl$_2$ and 100 μM ligand, 20 μM C8-PIP2), and 90 μM βarr1(ΔCT) was added. After incubation on ice for 1 h, the μOR was diluted to 1 nM in arrestin buffer with βarr1(ΔCT) at a final concentration of 90 μM. After immobilization, unbound μOR was washed out with imaging buffer supplemented with 90 μM βarr1(ΔCT) and movies were taken.

To extract the time trajectories of single-molecule fluorescence, collected movies were analysed by a custom-made software program developed as an ImageJ plugin (http://rsb.info.nih.gov/ij). Fluorescence spots were fitted by a 2D Gaussian function within a nine-pixel by nine-pixel area, matching the donor and acceptor spots using a variant of the Hough transform[55]. The background subtracted total volume of the 2D Gaussian peak was used as raw fluorescence intensity $I$.

Actual FRET efficiency was calculated via equation $E = \left(1 + \frac{I_D}{I_A - \chi I_D} \gamma\right)^{-1}$, where $I_D$ is raw fluorescence intensity of donor, $I_A$ is raw fluorescence intensity of acceptor, and $\chi$ is the cross-talk of the donor emission into the acceptor channel. $\gamma$ accounts for the differences in quantum yield and detection efficiency between the donor and the acceptor and is

calculated as the ratio of change in the acceptor intensity ($\Delta I_A$) to change in the donor intensity ($\Delta I_D$) upon acceptor photobleaching[56] ($\gamma = \Delta I_A / \Delta I_D$). The $\chi$ was 0.05, and the $\gamma$ was 1 and 0.2 for Cy3/Cy5 and Cy3/Cy7 dye pairs, respectively. FRET traces were picked by a custom-made Matlab script based on three criteria[57]: (1) signal-to-nose ratio of trances, which is defined as the mean of total intensity before photobleaching divided by its standard deviation, was higher than 4 and 3 for Cy3/Cy5 and Cy3/Cy7 dye pairs, respectively; (2) donor traces have single-step photobleaching; (3) traces last for at least 2 s. To calculate the transition rate in the presence of G protein and GDP, only traces that showed at least one high/low-FRET transition were selected and analysed by a Hidden Markov model-based software (HaMMy)[44]. Two FRET states were identified by HaMMy. The cumulative frequency count of high-FRET dwell times for each condition was fitted in Origin software to single exponential decay curves, generating high-FRET dwell time. The cumulative frequency count of low-FRET dwell times for each condition was fitted in Origin software to double exponential decay curves and the low-FRET dwell time was calculated as a weighted average accordingly.

### DEER experiments and analysis

**Setup.** Four pulse, Q-band DEER data were collected at 50 K on a Bruker e580 equipped with a QT-II resonator and a 150 W TWT amplifier using the pulse sequence: $\pi/2(\nu_A) - \tau_1 - \pi(\nu_A) - (\tau_1 + t) - \pi(\nu_B) - (\tau_2 - t) - \pi(\nu_A) - \tau_2 - $ echo, with $\tau_1 = 300$ ns, $\tau_2 = 3.5$ μs, $\Delta t = 16$ ns, 16-step phase cycling and a repetition time of 510 μs. The observer pulses ($\nu_A$) were set to 18 ns and 36 ns for $\pi/2$ and $\pi$ pulses, respectively, and applied 70 MHz below resonance. The 100 ns pump pulse ($\nu_B$) was applied on resonance and consisted of a 50 MHz linear chirp pulse generated by an arbitrary waveform generator. We furthermore used an 8-step ESEEM suppression protocol. All experiments were implemented using Xepr v2.6b.163.

**Analysis.** DEER data were processed via Gaussian mixture models (GMM) implemented in Matlab (v.2019b) using the DEERlab toolbox (v.0.9.2)[58]. In brief, all 30 datasets (10× ligand only, 10× ligand + G$_i$, 10× ligand + β-arr) were analysed simultaneously assuming a variable number of two to seven Gaussians whose mean positions and widths (global fitting parameters) were constrained in the range of 20–100 Å, and 2–20 Å, respectively. For each individual condition the sum of populations (local fitting parameters) was normalized to 1. Each of the thirty datasets was allowed a unique modulation depth (range 0.3–0.7) and each transducer condition allowed for a unique receptor concentration in the range of 25–150 μM. Model-based distance distributions and background corrected dipolar kernels were calculated using DEERlab functions and fit simultaneously to all 30 datasets using the fitparamodel.m routine (Multistart = 10). Post hoc model selection was performed using the Akaike information criterion corrected (AICc) and the more restrictive Bayesian information criterion (BIC) which were both evaluated globally for all DEER datasets and both yielded 6 Gaussians as most parsimonious model. Error analysis using 1,000 bootstrap iterations was performed for all fitting parameters, the dipolar fits and the parametric distance distributions, and evaluated at the 95% confidence level. Significant population changes between different transducer conditions were determined by disjunct 95% confidence intervals and are marked with * (star).

**Comparison of model-based and model-free analysis.** As a control, we also analysed all DEER data using Tikhonov regularization (TR) and model-free based analysis in DEERlab and LongDistances (v.946; http://www.biochemistry.ucla.edu/Faculty/Hubbell/software.html). Regularization or smoothness parameters were determined via AICc and L-curve criterion, respectively. The results from both analyses were superimposable. For comparison, the distance distributions derived

from the model-based (6 Gaussian) best fit and model-free DEERlab fits are shown in Extended Data Fig. 5. Both methods yield almost identical distance distributions and reveal all ligand or transducer-dependent distance changes supporting the validity of the model-based fit. Most apparent differences appear in the 35–45-Å distance range, where model-based analysis was able to differentiate two peaks, namely at 39 Å and 43 Å, of different width, namely 3.8 Å and 2 Å. This finding exemplifies one of the inherent advantages of the global, GMM-based fitting approach over Tikhonov regularization or model-free analysis. While Tikhonov regularization or model-free based analyses apply a single regularization or smoothness parameter to the full distance range, the chosen GMM allows different widths for individual distance peaks, as they may exist for different conformational states. Other advantages of the model-based approach include straightforward quantification of each population (Gaussian area) and a rigorous error analysis for each fitting parameter using covariance matrix or bootstrapping based approaches.

We conducted biological repeats for naloxone and lofentanil with and without G protein. These conditions represent the most distinct ligand/transducer conditions investigated and we observe good reproducibility. In particular, for both ligands, the smaller $G_i$-induced shifts are accurately reproduced (Extended Data Fig. 8d).

## Radioligand binding

Membranes of Sf9 cells expressing µOR were used for saturation binding and competition binding. Saturation binding was performed by incubating Sf9 membrane with increasing concentrations of the antagonist [$^3$H]diprenorphine ($^3$H-DPN, Perkin Elmer) for 2 h at room temperature in 0.5 ml of binding buffer containing 50 mM Tris-HCl pH 7.5, 100 mM NaCl, 0.1% BSA. Nonspecific binding of $^3$H-DPN was measured by adding 10 µM naloxone in the binding reaction. To separate unbound $^3$H-DPN, binding reactions were rapidly filtered over GF/B Brandel filters. The filters were then washed three times with 5 ml ice-cold binding buffer. Radioactivity was assayed by liquid scintillation counting.

For competition binding with $^3$H-DPN, Sf9 cell membrane was incubated with 2.9 nM $^3$H-DPN and increasing concentrations of DAMGO in 0.5 ml of binding buffer. Binding reactions were incubated for 2 h at room temperature. The free ligand was separated by rapid filtration onto a GF/B Brandel filter with the aid of a 48-well harvester (Brandel). Radioactivity was assayed by liquid scintillation counting.

The resulting data were analysed using Prism 9.0 (GraphPad Software). The dissociation constant ($K_d$) of $^3$H-DPN was calculated by fitting the saturation data in a one-site (total and nonspecific binding) model. The $K_i$ of DAMGO was calculated by fitting the competition binding data in a one-site (fit $K_i$) model.

For competition binding with [$^3$H]naloxone, mouse µOR-containing insect cell membranes prepared above were diluted to normalize expression levels between wild-type (1:1,000) and minimal-cysteine mouse µOR (1:100) in 20 mM HEPES pH 7.4, 100 mM NaCl, and 0.05% BSA. Membranes were then incubated with 3 nM [$^3$H]naloxone and serially-diluted orthosteric ligands at their respective final concentrations. Tested ligands were diluted into the buffer above to a final concentration of 100 µM with a fourfold serial dilution series for 10 total concentrations. The only exception is BU72, which was diluted to 1.3 µM final concentration before the same serial dilution. All ligands include independent 'no ligand' controls (100% binding) and excess cold naloxone (200 µM) controls (0% binding) to which points were normalized. The mixtures were shaken for 1 h at room temperature before collection onto Filtermat B (Perkin Elmer) and washed with cold binding buffer (20 mM HEPES pH 7.4, 100 mM NaCl). The filters were then dried at 60 °C before adding a sheet of MultiLex B/HS melt-on scintillator sheets (Perkin Elmer) and counts read on a MicroBeta Counter (Perkin Elmer). Quadruplicate data values were plotted and normalized as described above.

## BRET-based assays with TRUPATH and arrestin signalling

The BRET-based assays were based on TRUPATH[59] and arrestin signalling[48]. To measure µOR's coupling with $G_{i1}$, HEK 293 T cells (ATCC CRL-3216, authenticated by the supplier, routinely tested for mycoplasma) were plated in 10 cm dishes at 3–4 million cells per dish in Dulbecco's Modified Eagle's Medium (DMEM) supplemented with 10% FBS. The next day, cell medium was replaced with fresh DMEM + 10% FBS medium. Cells were transfected 2 h later, using a 1:1:1:1 DNA ratio of receptor:Gα-RLuc8:Gβ1:Gγ2-GFP2 (500 ng per construct). Transit 2020 (Mirus Biosciences) was used to complex the DNA at a ratio of 3 µl Transit per µg DNA, in OptiMEM (Gibco-ThermoFisher) at a concentration of 10 ng DNA per µl OptiMEM. The next day, cells were collected from the plate using Versene (0.1 M PBS + 0.5 mM EDTA, pH 7.4) and plated in poly-D-lysine-coated white, clear-bottom 96-well assay plates (Greiner Bio-One) at a density of 50,000 cells in 200 µl culture medium (DMEM + 1% dialysed FBS) per well. The next day, white backings (Perkin Elmer) were applied to the plate bottoms, and growth medium was carefully aspirated and replaced immediately with 60 µl of assay buffer (1× Hank's balanced salt solution (1× HBSS, Gibco), 20 mM HEPES, pH 7.4), supplemented with 5 µM (final concentration) coelenterazine 400a (Nanolight Technologies). After a 5 min equilibration period, cells were treated with 30 µl of drug (3×) prepared in assay buffer for an additional 5 min. Plates were then read in an LB940 Mithras plate reader (Berthold Technologies) with 395 nm (RLuc8-coelenterazine 400a) and 510 nm (GFP2) emission filters, at integration times of 1 s per well. Plates were read serially four times, and measurements from the fourth read were used in all analyses. BRET ratios were computed as the ratio of the GFP2 emission to RLuc8 emission.

To measure coupling of µOR coupling with β-arrestin-1, the procedures are mostly similar to those in BRET-G-protein assays except: HEK 293 T cells were co-transfected in a 1:5 ratio with µOR-Rluc8 and Venus–β-arrestin-1. Before the addition of tested drugs, white backings (Perkin Elmer) were applied to the plate bottoms, and growth medium was carefully aspirated and replaced immediately with 60 µl of assay buffer (1× HBSS, 20 mM HEPES, pH 7.4), supplemented with 5 µM (final concentration in assay buffer) coelenterazine h (Nanolight Technologies). After a 5 min equilibration period, cells were treated with 30 µl of drug (3×) prepared in assay buffer for an additional 5 min. Plates were then read in an LB940 Mithras plate reader (Berthold Technologies) with 485 nm (RLuc8-coelenterazine h) and 530 nm (Venus) emission filters, at integration times of 1 s per well. Plates were read serially four times, and measurements from the fourth read were used in all analyses. BRET ratios were computed as the ratio of the Venus emission to RLuc8 emission. The BRET ratio from G-protein or arrestin assays was plotted using nonlinear regression and Dose-response stimulation equation in Prism 9 (Graphpad).

## Reporting summary

Further information on research design is available in the Nature Portfolio Reporting Summary linked to this article.

## Data availability

All data are available in the manuscript or supplementary materials, or from the corresponding authors upon reasonable request. Raw DEER data are available at https://doi.org/10.5281/zenodo.10631251 (ref. 60). Materials described in this study are available upon a request sent to the corresponding authors. Source data are provided with this paper.

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

**Acknowledgements** The authors thank T. Kálai for providing the HO-1427 spin label reagent; S. Peng for assistance with smFRET data collection and analysis; and D. Hilger for assistance with cwEPR measurements and helpful discussion. Support for this study came from the following funding sources: the National Natural Science Foundation of China (Grants 21922704, 22061160466, and 22277063 to C.C.), the National Institutes of Health (Grants R01GM137081 to M.E. and R37DA036246 to B.K.K.), the Deutsche Forschungsgemeinschaft (DFG, Grant 421152132 to M.E.), the National Research Development and Innovation Office (Grant NKFI 137793 to C.P.S.), the Beijing Frontier Research Center for Biological Structure (to C.C.) and the Jules Stein endowed chair (to W.L.H.).

**Author contributions** J.Z. designed and validated constructs of the μOR, performed radioligand binding assays, purified all the proteins, performed phosphorylation of the μOR, synthesized iodoacetamide Cy3 and Cy5, labelled the μOR for smFRET and DEER studies, collected and analysed smFRET data and wrote the manuscript. M.E. performed DEER experiments, data analysis and wrote the manuscript. E.S.O. prepared DEER samples with J.Z. C.P.S. designed HO-1427. A.E.D. conducted cell signalling assays. J.H. assisted with smFRET data collection and analysis. X.S. assisted with protein purification. E.W. performed radioligand binding assays. T.C. supervised the cell signalling assays and analysis. W.L.H. supervised DEER data collection and analysis. B.K.K. and C.C. supervised the overall project and wrote the manuscript. All authors contributed to the manuscript preparation.

**Competing interests** B.K.K. is a cofounder of and consultant for ConfometRx. The other authors declare no competing interests.

**Additional information**
**Correspondence and requests for materials** should be addressed to Matthias Elgeti, Brian K. Kobilka or Chunlai Chen.

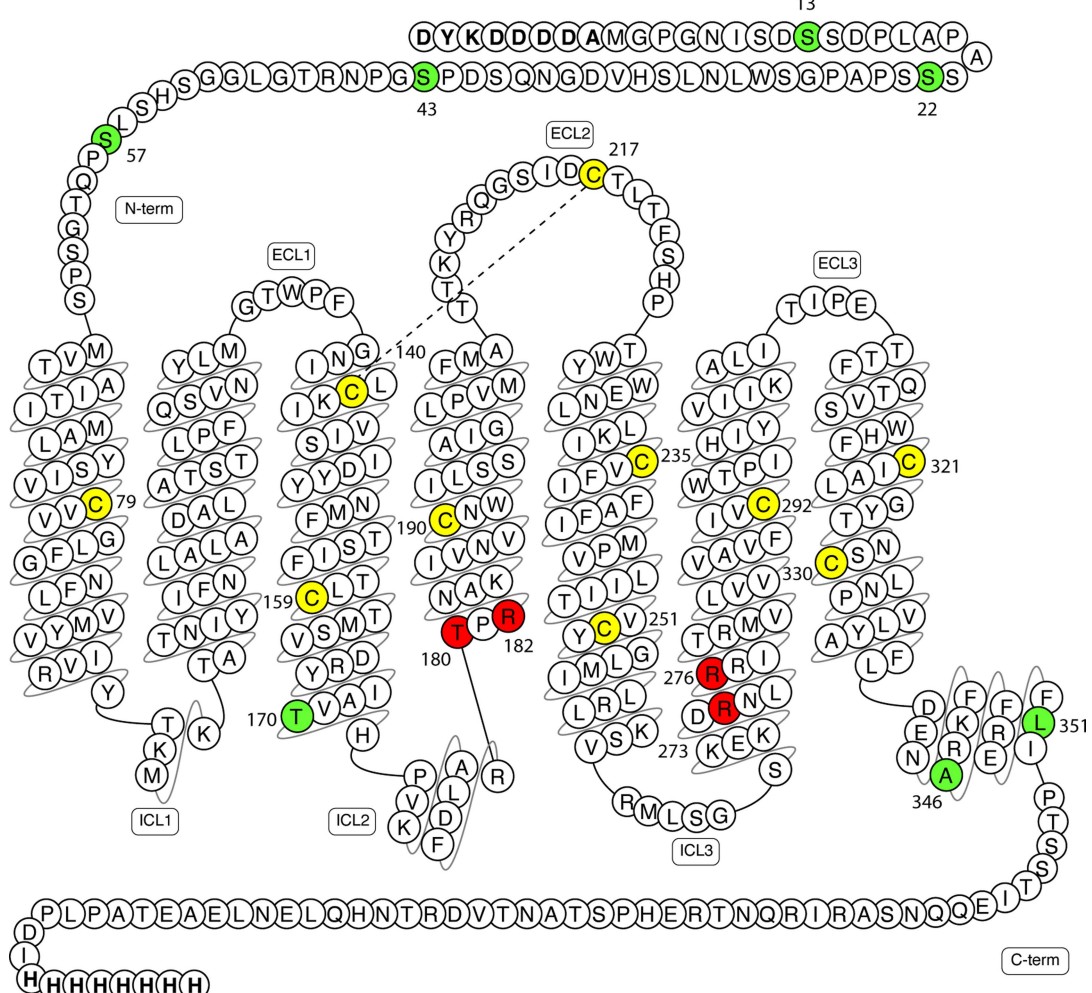

**Extended Data Fig. 1 | Snake plot of the μOR sequence.** Cysteine residues in yellow indicate native cysteine that were kept in the labeling constructs. Residues in green indicate native cysteine that is mutated to corresponding residues to avoid nonspecific labeling by cysteine reactive labeling reagent. Residues in red indicate labeling sites that are mutated to cysteine for labeling with cysteine-reactive fluorophores and nitroxide spin labels.

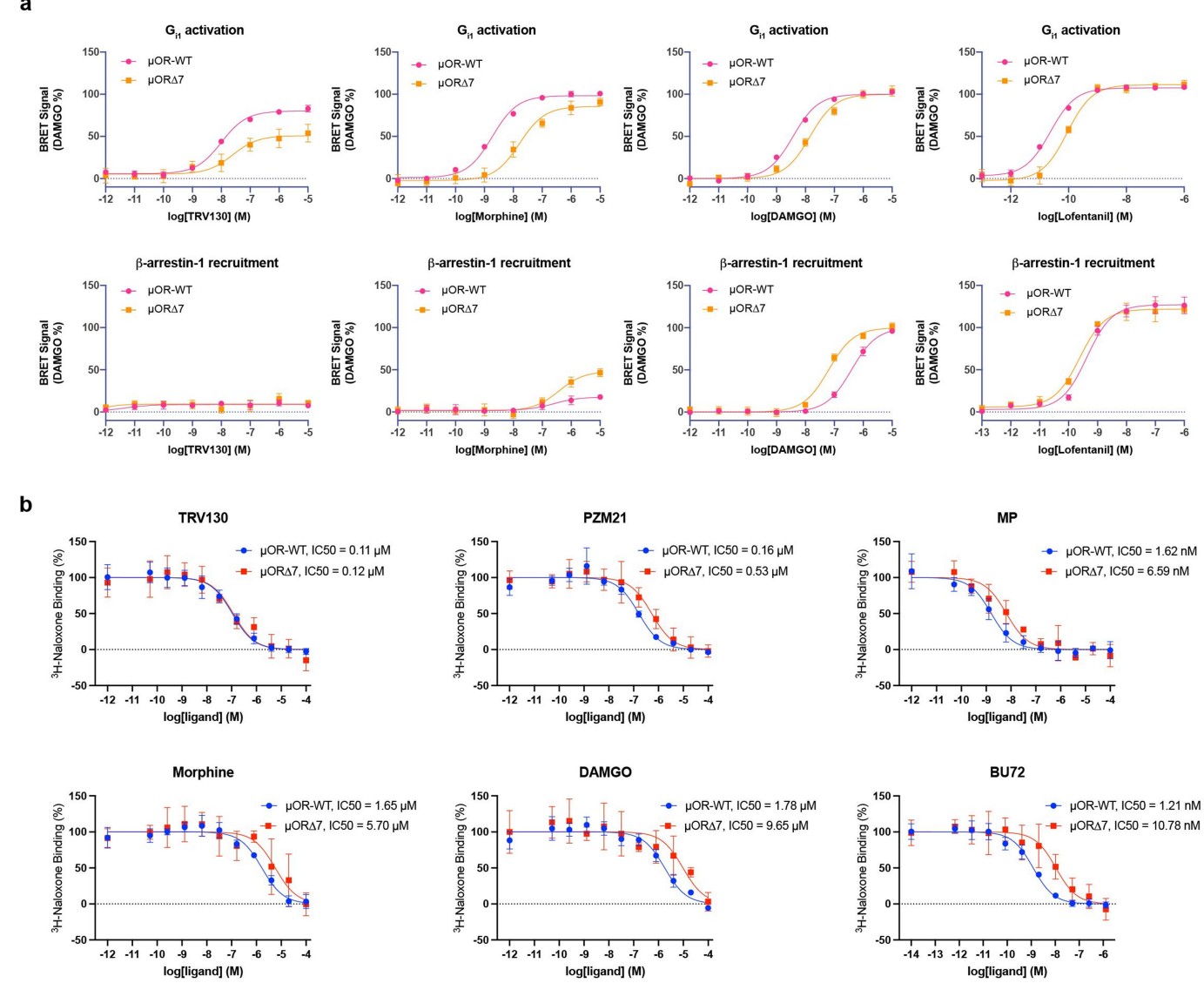

**Extended Data Fig. 2 | Function validation of minimal-cysteine μOR (μORΔ7). a**, Ligand efficacy of μORΔ7 is determined by BRET-based assays. Error bars represent s.d. from n = 6 samples. **b**, Ligand-binding affinity of μORΔ7 is determined by radioligand binding assay. The cell membrane of sf9 cells expressing μOR-WT or μORΔ7 was extracted and used for competition binding. The hot ligand is ³H-naloxone. Error bars represent s.d. from n = 4 samples.

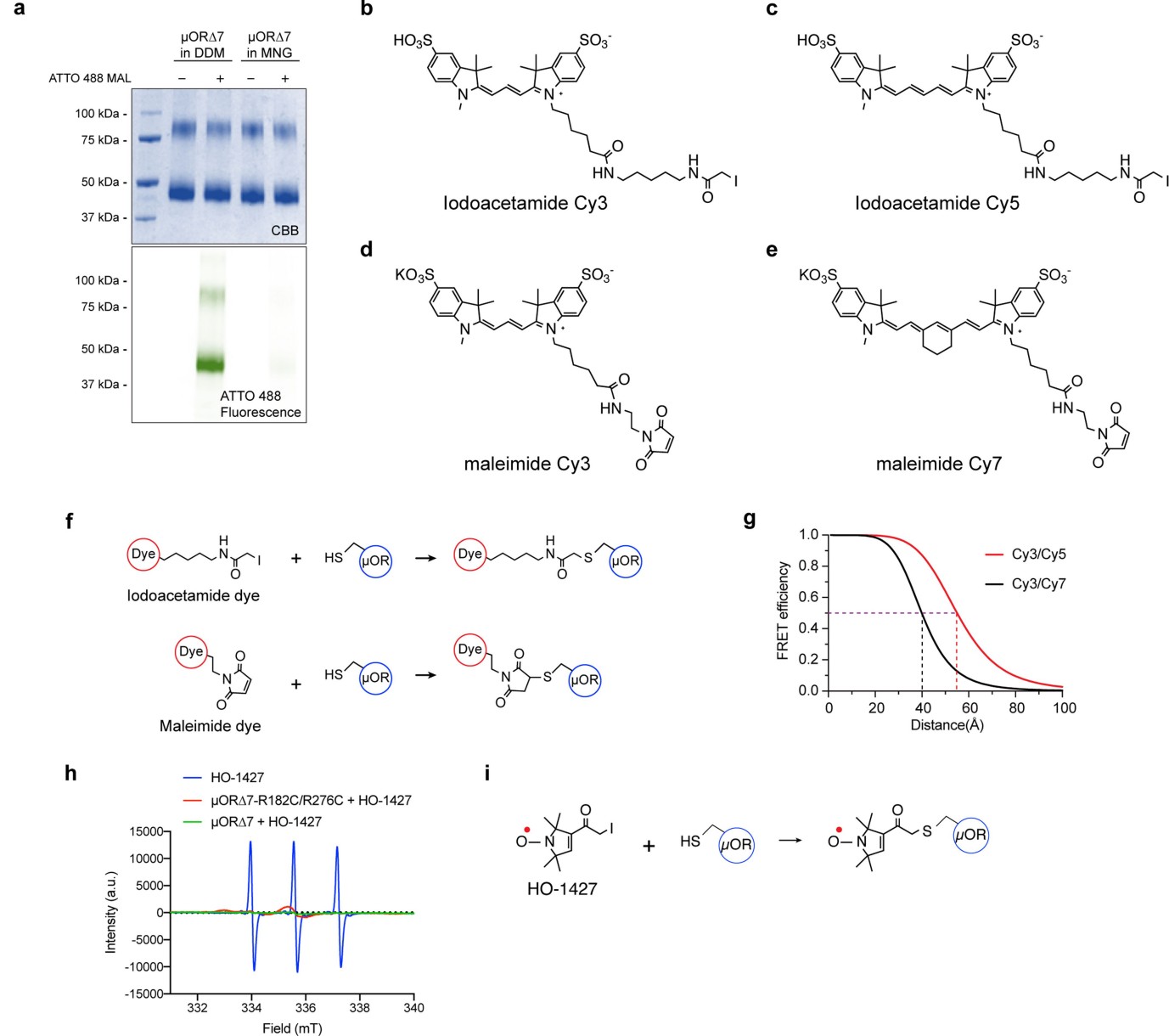

**Extended Data Fig. 3 | Minimal cysteine μOR (μORΔ7) shows minor nonspecific labeling by fluorophore or nitroxide spin label. a,** μORΔ7 purified in LMNG detergent shows almost no labeling by maleimide ATTO 488 as compared to μORΔ7 in DDM. 3 biological repeats were examined by UV-Vis spectrophotometry and showed similar labeling behavior. For demonstration, one sample was further examined through the gel. CBB, Coomassie Brilliant Blue staining. For gel source data, see Supplementary Fig. 8. **b-c,** Structures of iodoacetamide Cy3 (b) and Cy5 (c) that were made in-house using NHS-Cy3 and Cy5 from Lumiprobe. **d-e,** Structures of maleimide Cy3 (d) and Cy7 (e) that are commercially available from Lumiprobe. **f,** Labeling reactions of the μOR by cysteine-reactive iodoacetamide dye or maleimide dye. **g,** FRET efficiencies of Cy3/Cy5 and Cy3/Cy7 pairs as a function of inter-dye distances calculated based on R0 values, 55 Å and 40 Å for Cy3/Cy5 and Cy3/Cy7 pairs, respectively. **h,** Continuous-wave Electron Paramagnetic Resonance (CW-EPR) spectrum for HO-1427 labeled μORΔ7 (green) shows minimal labeling as compared to the same amount of free HO-1427 (blue) and HO-1427 labeled μORΔ7-R182C/R276C (red). **i,** Labeling reaction of the μOR by of HO-1427.

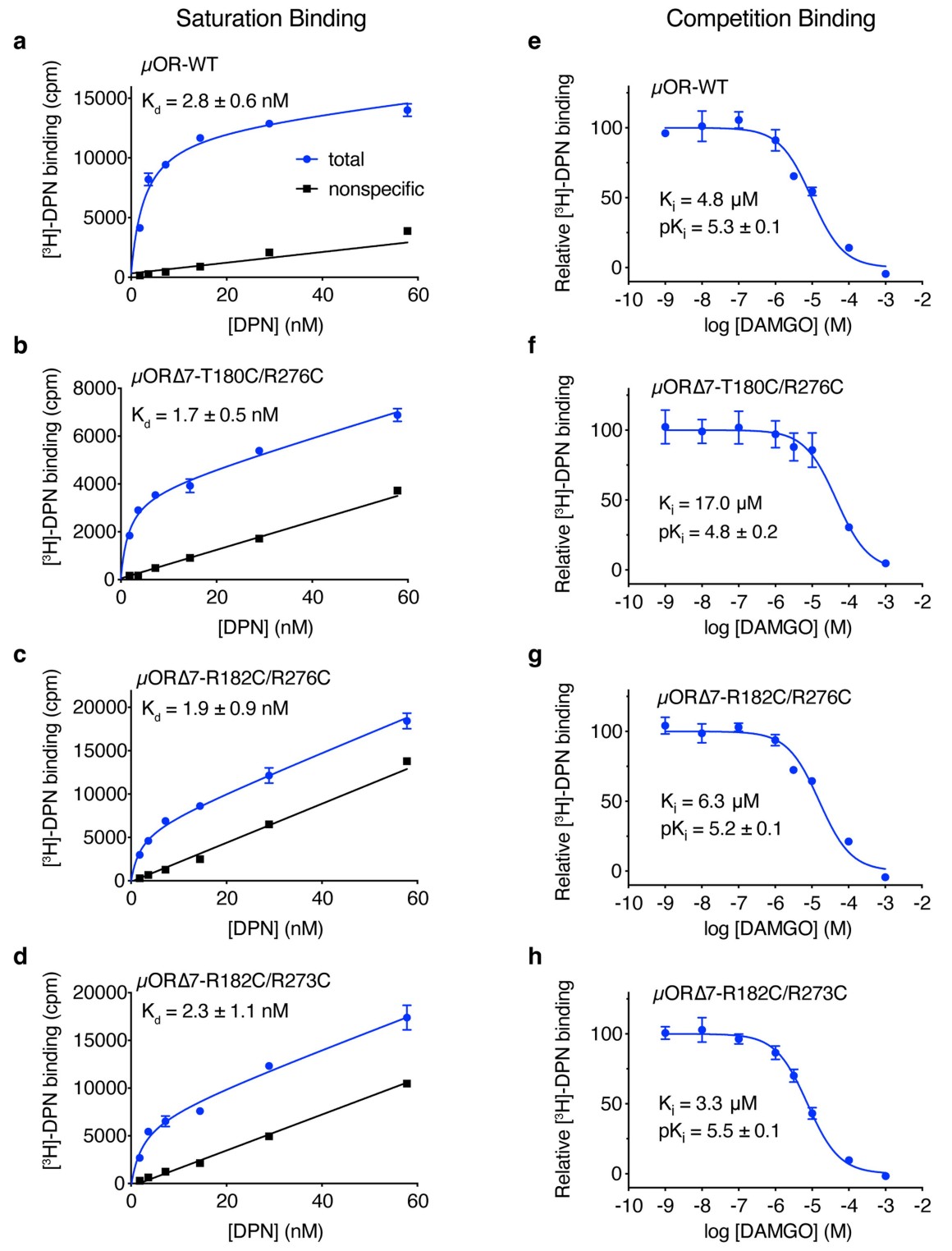

**Extended Data Fig. 4 | Radioligand binding of wild-type μOR (μOR-WT) and labeling constructs using sf9 insect cell membranes. a–d**, Saturation binding. **e–h**, Competition binding. Error bars represent the s.e.m from triplicate measurements. ± indicates s.e.

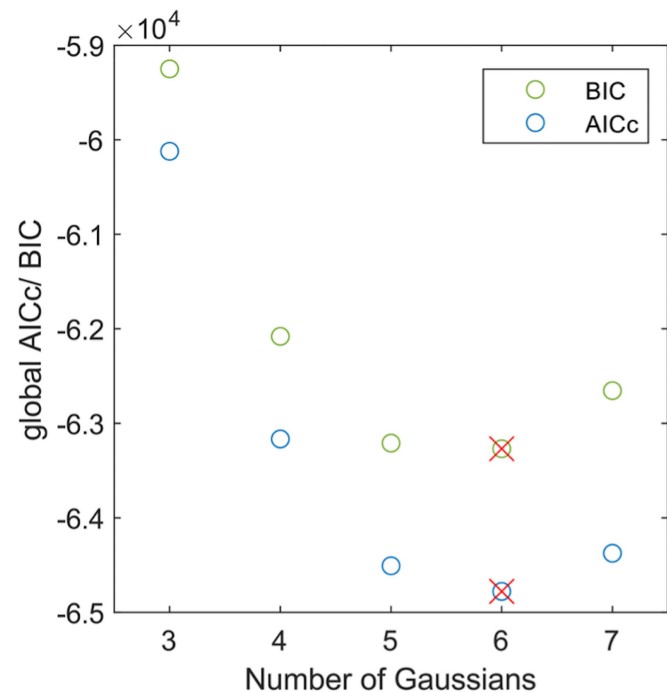

**Extended Data Fig. 5 | Gaussian model selection for DEER was based on global AICc and BIC values.** Both AICc and BIC values yield minimum values for six Gaussians (red cross) which was chosen as most parsimonious model.

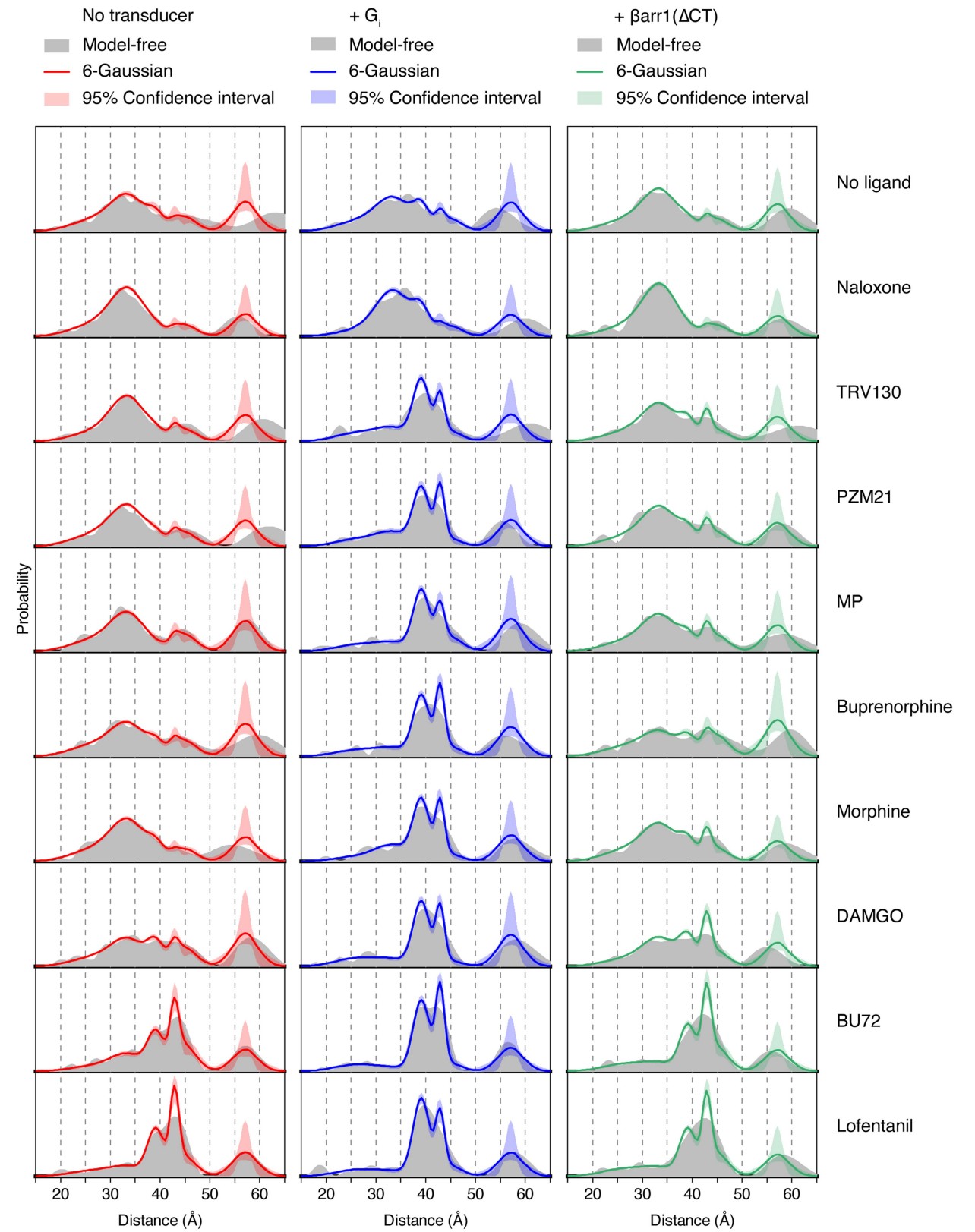

**Extended Data Fig. 6 | DEER distance distributions of the μOR using a model-free analysis vs the 6-Gaussian global fitting.**

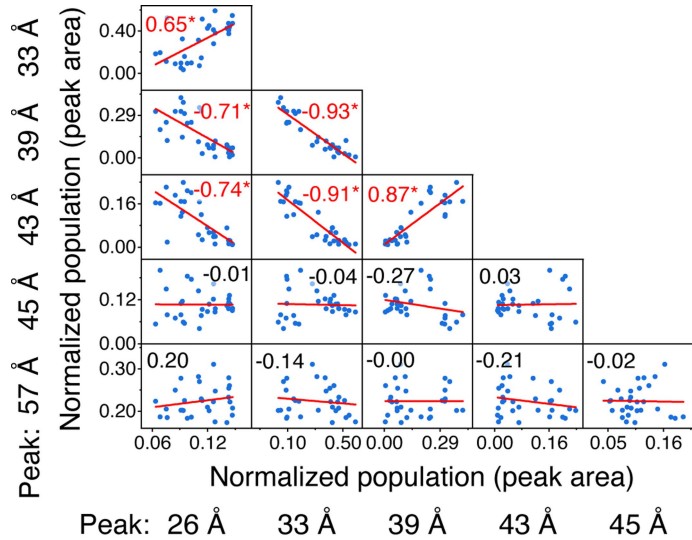

**Extended Data Fig. 7 | Correlation analysis of DEER populations.**
Populations from 6 Gaussian peaks of 30 DEER datasets are shown as scatter plot. Each blue dot represents one of the 30 samples. Red lines are the results of a linear fit. Numbers in each subpanel are corresponding correlation coefficients, which are labeled by a star (*) and red color if $p < 0.05$.

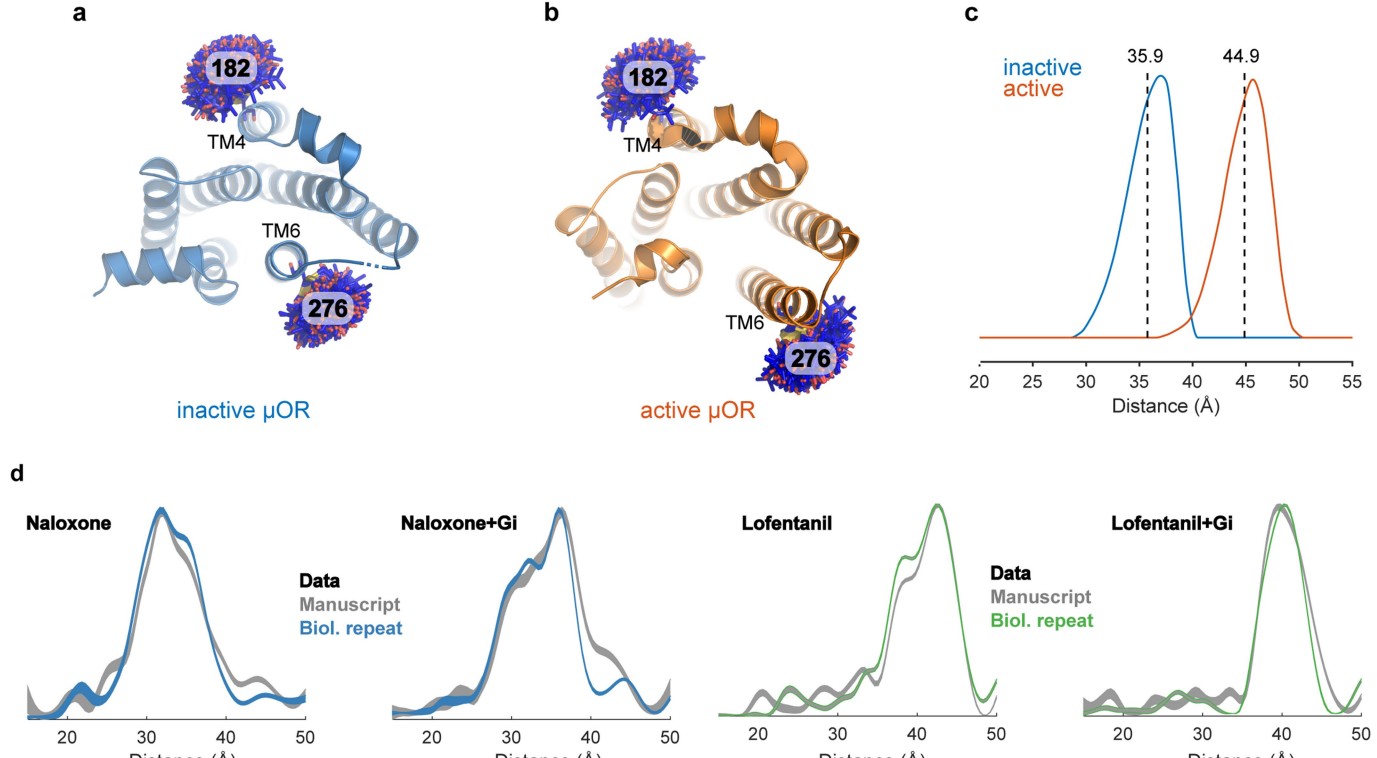

**Extended Data Fig. 8 | Modeling of spin-labeled μOR. a,b,** HO-1427 labeled μOR was modeled using mtsslWizard and the inactive (a) and active (b) μOR structures. **c,** Distance distributions between 182 in TM4 and 276 in TM6 derived from (a) and (b). Dashed lines indicate average distances. **d,** Biological repeats for four selected ligand and transducer conditions.

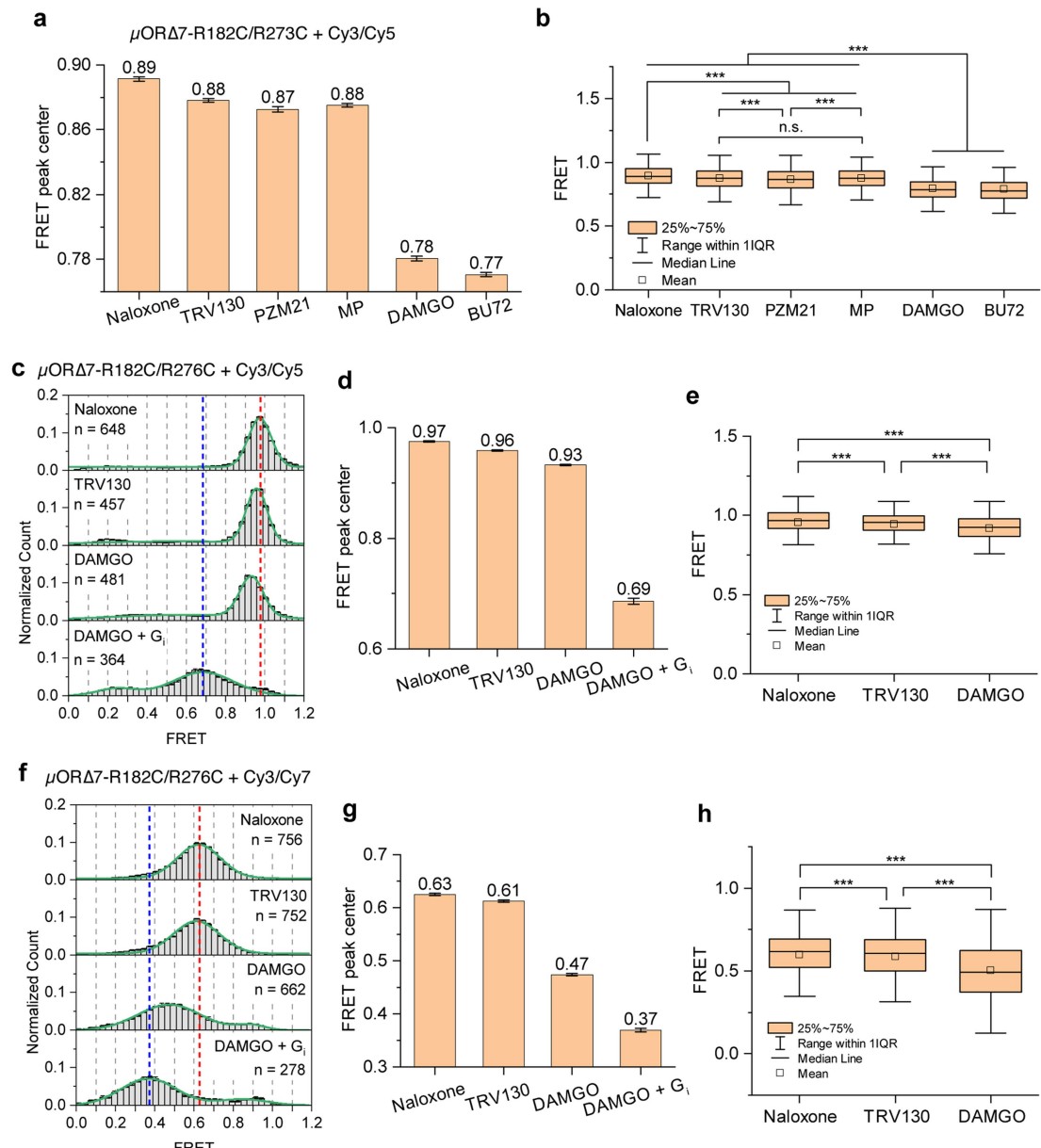

**Extended Data Fig. 9 | FRET peak centers and average FRET values of fluorophore-labeled μOR. a,b,** μORΔ7-R182C/R273C is labeled with Cy3/Cy5. **c–e,** μORΔ7-R182C/R276C is labeled with Cy3/Cy5. **f–h,** μORΔ7-R182C/R276C is labeled with Cy3/Cy7. FRET distributions of μORΔ7-R182C/R276C labeled with Cy3/Cy5 (c) and Cy3/Cy7 (f). Error bars in c and f indicate s.d. from 3 repeats. FRET peak centers of μORΔ7-R182/R273 + Cy3/Cy5 (a, related to Fig. 3b), μORΔ7-R182/R276 + Cy3/Cy5 (d), and μORΔ7-R182/R276 + Cy3/Cy7 (g). The numbers on each bar are the peak centers extracted from the Gaussian fitting. Error bars indicate standard errors of the fitting. FRET values of each frame of μOR samples in Fig. 3b (b), Extended Data Fig. 9c (e), and Extended Data Fig. 9f (h) are plotted as box-and-whisker plots. IQR, inter qaurtile range. The number of traces of each condition is indicated in the corresponding histograms. FRET efficiencies between 0.6 and 1.2 (b and e) and between 0 and 1.2 (h) were used for one-way ANOVA Tukey's test. ***, p < 0.001. n.s., not significant.

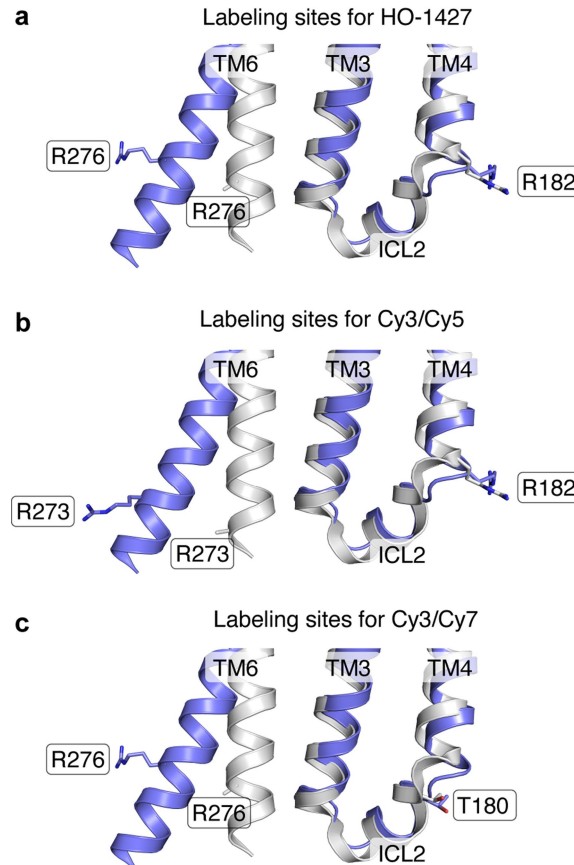

**a** Labeling sites for HO-1427

TM6 TM3 TM4

R276 R276 R182 ICL2

**b** Labeling sites for Cy3/Cy5

TM6 TM3 TM4

R273 R273 R182 ICL2

**c** Labeling sites for Cy3/Cy7

TM6 TM3 TM4

R276 R276 T180 ICL2

**Extended Data Fig. 10 | Labeling sites of µOR for nitroxide spin HO-1427 and different fluorophore pairs.** Inactive µOR structure (in grey, PDB code 4DKL) and G protein-coupled active µOR structure (in blue, PDB code 6DDF) are superimposed. Transmembrane 1, 2, 5, and 7 are hidden for clarity. **a**, Arg182C and Arg276C were labeled by HO-1427. **b**, Arg182C and Arg273C were labeled by Cy3/Cy5 pair. **c**, Tyr180C and Arg276C were labeled by Cy3/Cy7 pair. Sidechains of Arg273 and Arg276 were not modeled in the published inactive structure (PDB 4DKL).

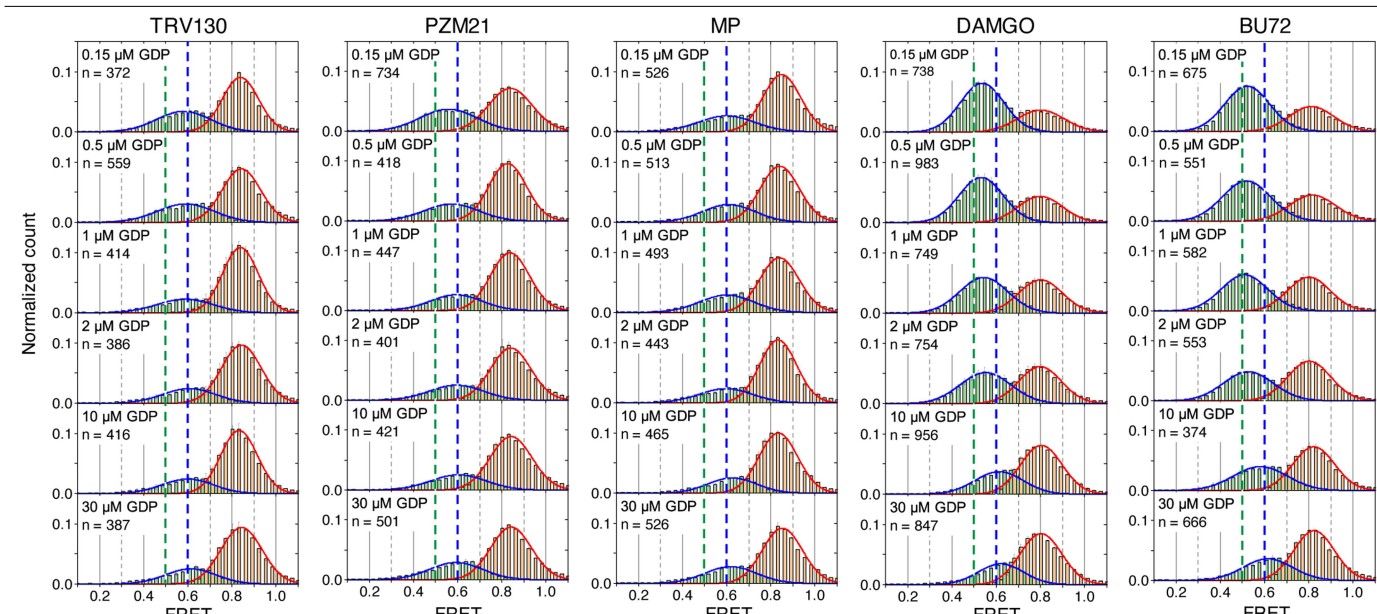

**Extended Data Fig. 11 | FRET histogram of high and low FRET states.** FRET traces in the presence of 20 µM Gi, increasing concentrations of GDP, and ligand of TRV130, PZM21, MP, DAMGO, or BU72 were analyzed using a two-state hidden Markov Model. Only traces with at least one transition were selected. Frames of high-FRET and low-FRET states were extracted separately and binned to plot histograms. FRET histograms of high FRET (bars in orange) and low FRET (bars in green) states are shown and fitted to Gaussians (solid curves in red and blue, respectively). FRET efficiencies at 0.5 and 0.6 are highlighted with dashed lines in green and blue, respectively. n, number of fluorescence traces used to calculate the corresponding histograms. Error bars represent s.e.m. from 2 repeats.

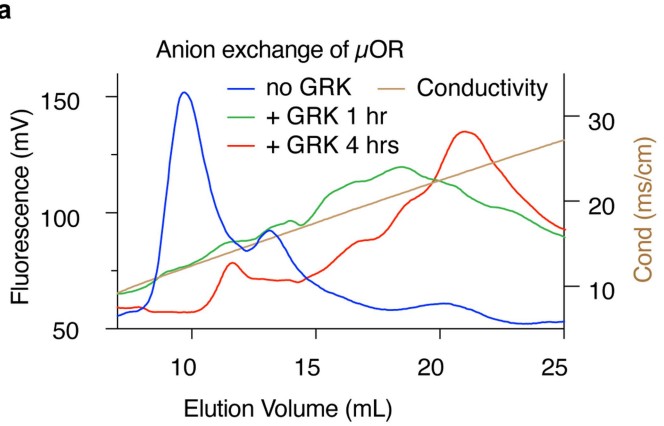

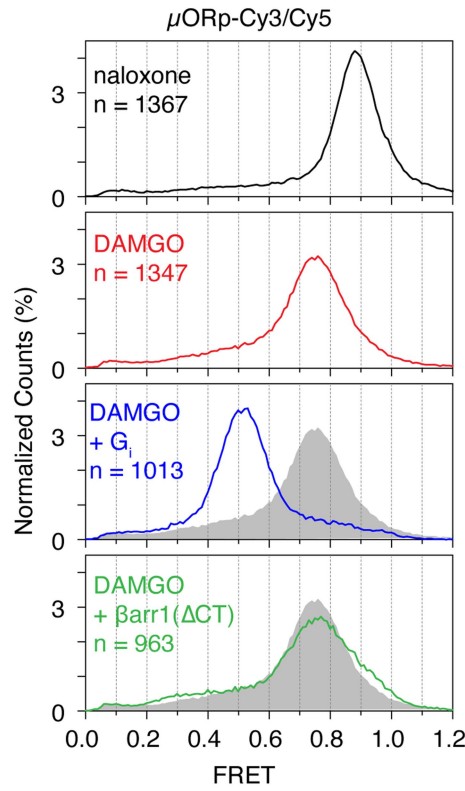

**Extended Data Fig. 12 | Effect of β-arrestin-1 on smFRET distribution of phosphorylated μOR-Cy3/Cy5. a**, Phosphorylation of μORΔ7-182 C/276 C by GRK5 for DEER spectroscopy. Anion exchange chromatography (MonoQ) was used to find the best condition for GRK5 phosphorylation of the μOR. **b**, smFRET distributions of GRK5-phosphorylated μORΔ7-182 C/273C-Cy3/Cy5 (μORp-Cy3/Cy5) in the presence of 20 μM of C8-PIP2. The grey shaded areas are the DAMGO alone condition for comparison.

|  | Corresponding author(s): | Matthias Elgeti, Brian K. Kobilka and Chunlai Chen |
| --- | --- | --- |
|  | Last updated by author(s): | 02/15/2024 |

# Reporting Summary

## Statistics

For all statistical analyses, confirm that the following items are present in the figure legend, table legend, main text, or Methods section.

| n/a | Confirmed |  |
| --- | --- | --- |
| ☐ | ☒ | The exact sample size (*n*) for each experimental group/condition, given as a discrete number and unit of measurement |
| ☐ | ☒ | A statement on whether measurements were taken from distinct samples or whether the same sample was measured repeatedly |
| ☐ | ☒ | The statistical test(s) used AND whether they are one- or two-sided<br>*Only common tests should be described solely by name; describe more complex techniques in the Methods section.* |
| ☒ | ☐ | A description of all covariates tested |
| ☐ | ☒ | A description of any assumptions or corrections, such as tests of normality and adjustment for multiple comparisons |
| ☐ | ☒ | A full description of the statistical parameters including central tendency (e.g. means) or other basic estimates (e.g. regression coefficient) AND variation (e.g. standard deviation) or associated estimates of uncertainty (e.g. confidence intervals) |
| ☐ | ☒ | For null hypothesis testing, the test statistic (e.g. *F*, *t*, *r*) with confidence intervals, effect sizes, degrees of freedom and *P* value noted<br>*Give P values as exact values whenever suitable.* |
| ☒ | ☐ | For Bayesian analysis, information on the choice of priors and Markov chain Monte Carlo settings |
| ☒ | ☐ | For hierarchical and complex designs, identification of the appropriate level for tests and full reporting of outcomes |
| ☒ | ☐ | Estimates of effect sizes (e.g. Cohen's *d*, Pearson's *r*), indicating how they were calculated |

*Our web collection on statistics for biologists contains articles on many of the points above.*

## Software and code

Policy information about availability of computer code

| Data collection | Single-molecule fluorescence data was collected using Cell Vision software V1.4.0 (Beijing Coolight Technology), DEER data was collected in Xepr v2.6b.163 (Bruker, Ettlingen, Germany) |
| --- | --- |
| Data analysis | We used the following published software and web based analysis tools to analysis our data as referenced in methods: Origin 9.8.0.200 (OriginLab); ImageJ v1.43u; Matlab R2017a; HaMMy v4.0; DEERlab v0.9.2; LongDistances v.946; |

For manuscripts utilizing custom algorithms or software that are central to the research but not yet described in published literature, software must be made available to editors and reviewers. We strongly encourage code deposition in a community repository (e.g. GitHub). See the Nature Portfolio guidelines for submitting code & software for further information.

## Data

Policy information about availability of data

All manuscripts must include a data availability statement. This statement should provide the following information, where applicable:
- Accession codes, unique identifiers, or web links for publicly available datasets
- A description of any restrictions on data availability
- For clinical datasets or third party data, please ensure that the statement adheres to our policy

All the data are available in the manuscript or supplementary materials, or from the corresponding authors upon reasonable request. Raw DEER data is available at

# Human research participants

Policy information about studies involving human research participants and Sex and Gender in Research.

| | |
|---|---|
| Reporting on sex and gender | N/A |
| Population characteristics | N/A |
| Recruitment | N/A |
| Ethics oversight | N/A |

Note that full information on the approval of the study protocol must also be provided in the manuscript.

# Field-specific reporting

Please select the one below that is the best fit for your research. If you are not sure, read the appropriate sections before making your selection.

☒ Life sciences    ☐ Behavioural & social sciences    ☐ Ecological, evolutionary & environmental sciences

For a reference copy of the document with all sections, see nature.com/documents/nr-reporting-summary-flat.pdf

# Life sciences study design

All studies must disclose on these points even when the disclosure is negative.

| | |
|---|---|
| Sample size | Sample size for TIRF based fluorescence imaging was the number of individual molecules. Therefore, in most conditions, approximately, >300 individual molecules were analyzed, which are sufficient for statistical analysis. The fluorescence experiments were repeated 3+ times to allow calculation of the mean and standard error of the mean. For functional assays (such as BRET1 and BRET2), there are at least three biological replicates that are reported in the figure legends.<br>No sample size calculation was performed. The sample sizes are sufficient since each experiment was carried out with controls and replicated more than once. |
| Data exclusions | No data were excluded from the analysis. |
| Replication | All smFRET results were successfully reproduced through at least 2-4 independent attempts.<br>DEER data was collected for several different constructs and different spin labels using a limited set of conditions (Apo, naloxone, MP, morphine, PZM21, DAMGO, lofentanil, DAMGO + Gi). After optimizing construct and label, we acquired the full set of conditions reported here. For functional assay, data were replicated using three biological replicates. See figure legends for specific details. |
| Randomization | Not relevant to this study. |
| Blinding | Blinding was not possible, however, appropriate controls were used during experiments. |

# Reporting for specific materials, systems and methods

We require information from authors about some types of materials, experimental systems and methods used in many studies. Here, indicate whether each material, system or method listed is relevant to your study. If you are not sure if a list item applies to your research, read the appropriate section before selecting a response.

### Materials & experimental systems

| n/a | Involved in the study |
|---|---|
| ☐ | ☒ Antibodies |
| ☐ | ☒ Eukaryotic cell lines |
| ☒ | ☐ Palaeontology and archaeology |
| ☒ | ☐ Animals and other organisms |
| ☒ | ☐ Clinical data |
| ☒ | ☐ Dual use research of concern |

### Methods

| n/a | Involved in the study |
|---|---|
| ☒ | ☐ ChIP-seq |
| ☒ | ☐ Flow cytometry |
| ☒ | ☐ MRI-based neuroimaging |

## Antibodies

| | |
|---|---|
| Antibodies used | Monoclonal (clone M1) ANTI-FLAG antibody produced in mouse (Sigma, Cat. No. F3040). The antibody was not directly used, but was processed to generate Fab, which was then biotinylated. |
| Validation | Antibody was validated by the commercial supplier. The statement can be found on the website: https://www.sigmaaldrich.com/US/en/product/sigma/f3040 |

## Eukaryotic cell lines

Policy information about cell lines and Sex and Gender in Research

| | |
|---|---|
| Cell line source(s) | Sf9 and Hi5 cell lines were from Expression Systems, LLC (USA). Source for HEK 293T cell line is ATCC. |
| Authentication | The cell lines were not authenticated. |
| Mycoplasma contamination | The cell lines were not tested for mycoplasma contamination. |
| Commonly misidentified lines (See ICLAC register) | No commonly misidentified cell lines were used. |

