## [Peer Review File · Nature]

Manuscript Title: Ligand efficacy modulates conformational dynamics of the μ -opioid receptor

Reviewer Comments & Author Rebuttals

Reviewer Reports on the Initial Version:

Referee expertise:

Referee #1: GPCRs, including single molecule methods

Referee #2: EPR/DEER

Referee #3: opioid receptor pharmacology

Referees' comments:

Referee #1 (Remarks to the Author):

Review of

Jiawei Zhao, Matthias Elgeti et al.

Conformational dynamics of the μ -opioid receptor determine ligand intrinsic efficacy

The teams of Chunlai Chen, Brian Kobilka and Wayne Hubbell have combined efforts in order to delineate the dynamics of a G-protein-coupled receptor (GPCR). Using two highly sensitive methods, double electron-electron resonance (DEER) and single-molecule fluorescence resonance energy transfer (smFRET), the authors explore conformations of the μ -opioid receptor - a prototypical class A GPCR that mediates most of the wanted as well as unwanted effects of opioids. This receptor is interesting not only because it is of great clinical importance and has been studied in quite some detail, but also - in the context of the present work - because ligands with very different properties (different chemical nature, partial vs. full agonism, biased activation etc.)

The background of this work is formed by published X-ray and cryo-EM structures of the receptor alone or with its cognate G protein, G_i . These structures, however, most likely represent only stable states that may or may not reflect the most relevant conformations. In addition, many more conformations may exist that are not stable enough to be amenable to crystallization or cryo-EM.

Although there are variations in details between the constructs used in the different experiments, the authors concentrate on a single axis in the receptor, i.e. TM4 - TM6. While this is logical based on the fact that the outward movement of TM6 is a hallmark of GPCR activation, it means that conformational heterogeneity is limited to a 2-dimensional view.

DEER studies could be resolved into 6 different peaks, of which 4 were deemed functionally relevant. Ligands produced only minor changes in these populations, with the exception of the super-agonists lofentanil and BU72, which led to marked increases in the long distance populations, compatible with the outward movement in TM6 during activation. In the additional presence of G_i , such an increase in the long distance populations became visible for all other ligands (with the obvious exception of the antagonist naloxone), suggesting that that G_i stabilized an active receptor conformation. Interestingly, the this active conformation was characterized for most ligands with roughly equal proportions of the 39Å and the 43Å populations (suggesting that the active G-protein-coupled state is composed of a mixture of these two states), while lofentanil and BU72 alone had led to a clear preponderance of the 43Å population. Also of interest is the

observation that Gi alone had little effect in these experiments, indicating that while it facilitated formation of an active state, it did not do so on its own.

In contrast to the major effects of Gi on these different receptor populations, the effects of beta-arrestin1 were more modest (compared to the respective ligands alone). This suggests that the effects of beta-arrestins on receptor conformations are much less pronounced than those of the G-protein(s).

smFRET experiments were then conducted to study the kinetics of interconversion between different receptor populations. Unfortunately, this method is also limited in temporal resolution, in this case to 100 ms, which leads to averaging between rapidly (i.e. <100ms) exchanging conformations. As a consequence, experiments with Cy3/Cy5 labelled receptors revealed only a shift in mean FRET that was observed with DAMGO and BU72 (low FRET indicating again outward movement of TM6) - roughly in line with the DEER data for superagonists and suggesting the presence of rapidly achieved active state(s). With the Cy3/Cy7 construct, an (additional) slower (i.e. >100ms) conformational change could be resolved. Such fast vs. slow conformational changes would be compatible with receptor activation kinetics observed in intact cells, where receptor activation itself occurs with time constants <100ms, whereas G-protein-dependent effects occur at time scales >100ms.

smFRET experiments at different concentrations of GDP led the authors to two main kinetic conclusions: (a) super-agonists increase the speed of binding of Gi to the receptor, and (b) G-protein-biased ligands do not lower GDP-affinity for Gi. Finding (a) is compatible with data from receptor/Gi-interactions in intact cells, where agonists have been reported to increase the kon of binding. I find it hard to understand finding (b), because I would assume that even G-protein-biased ligands would need to induce GDP-release from Gi; how would this happen if the affinity is not lowered?

As in DEER experiments, no major specific effects of beta-arrestin1 were observed, confirming the notion of a less tight (and perhaps specific) nature of this interaction. It might be interesting to assess in these experiments of in DEER studies, whether the same would be true for beta-arrestin2, which in several instances has been reported to bind receptors more tightly.

Overall, this is a very well conducted, carefully performed and interpreted study by presumably the best team of authors that might address these issues. Aside from the considerations above I have no technical criticisms. I think, three aspects might be improved in the presentation/discussion: (1) For many aspects there is more data in the literature to support the authors' conclusions; a more encompassing discussion of this literature might help the authors to make their points, and might help the reader to position the present manuscript; this refers to both, experiments with isolated reconstituted systems including smFRET and experiments in intact cells; (2) it would be good if the authors might dare to move a little further than just interpret their data in terms of TM4-TM6 distances; what do we learn about the 3D cytosolic interface of the receptor in these different states? (3) it would be helpful if the authors could go a little further in explaining what they learn from their experiments beyond what is known from earlier NMR experiments, which have shown already the dynamic nature of inactive and active receptor conformations, the presence of multiple conformations that contrast with simple on/off-models etc.

Referee #2 (Remarks to the Author):

A: The authors report in their manuscript on a DEER smFRET study on the conformational states of the μ -opioid receptor upon binding of several different pharmaceutically relevant ligands in the presence and absence of two transducers, i.e., G protein Gi and β -arrestin-1). Their data are supported by ligand binding essays. They identify several new states and show that these interconvert on largely different time scales. They identify the molecular basis of differences in the action of the ligands and the influence of the ligands on the transducer action.

B: The combination of DEER and smFRET is highly interesting but not novel (see paper from Hagelueken et al. Nat Commun 2022). The application to the μ -opioid receptor is new, and the system as such highly relevant. The insight gained here is in my opinion a quantum leap in the understanding of this receptor and for the action of pharmaceuticals in general. The found molecular principles

C: The combination of DEER as an ensemble technique that yields the distribution of the whole ensemble and smFRET providing time scales and real time movements is highly appropriate for the question asked. Indeed, only the combination of the three methods (DEER, smFRET, and biochemical assays) made it possible to draw the conclusions that "TM6 movement alone does not define receptor activity". The data and their presentation is of high quality.

D: The data statistics is valid as performed, but:

1) The label HO-1427 is restricted in mobility, one should therefore expect orientation selective DEER time traces. Using a chirp pump pulse will reduce orientation selection but the detection pulses are still selective. Since some of the observed changes in the distance distributions are very subtle, it would need to be made sure that in none of the cases the onset or diminishing of orientation selection occurs e.g., by recording DEER time traces at different offsets from the pump pulse. If orientation selection is present this needs to be considered e.g., by summing up time traces at different offsets or by taking it explicitly into account in the analysis. This is crucial for the whole analysis.

2) The authors do not present repeats but argue that the whole data as such is consistent and represents as such the repeats. I do not agree with this argument because it is circular. Their analysis relies on the whole data set being correct. Repeats would also give a better idea about the error of the experiment. Also, according to the white paper of the community (J. Am. Chem. Soc. 2021, 143, 17875–17890) repeats must be presented (mixture of technical and biological ones).

3) Connected with 2), the authors find discrepancies between DEER and smFRET for DAMGO and G protein-biased agonists. The argument that a long flexible linker should increase a small structural change eludes me. Commonly, a small and rigid label can report on a small structural change but not a label that is connected via a long flexible linker to the site of the structural change. This is a major inconsistency. If the reason is the different label site, then this should be checked by labeling the same sites.

4) Connected with 2) and 3), the whole analysis is based on one label pair, this is not enough but must be complemented by additional pairs at least one better two.

5) The control experiment for checking whether the remaining Cysteins in the minimal cysteine mutant are labeled was done with Iodoacetamido proxyl while the labeling itself was then done with HO-1427. This control experiment must be repeated with the actual label.

E: At current stage the missing data leave for me a large question mark on the robustness, validity and reliability of the interpretation of the data and conclusions.

F: Fig. S11 please use the actual label (HO-1427) in mtsslWizard (the author of the program and host of the site is always willing to include new labels on request if he hasn't done so already).

G: 1) Give references for DEER and smFRET. I would suggest taking the two white papers of the respective communities. 2) Give a reference for mtsslWizard.

H: Clarity and context are fine. Two minor things: 1) Fig. S10. Caption: I suspect the authors mean $p < 0.5$ instead of 0.05. 2) The authors state that they did not include distances above 4.4 nm into the analysis, which is good. But when discussing the reasons please also include a statement that the lengths of the time traces exclude their analysis ($r_{\max} = 30 (t_{\max}/\mu\text{s})^{1/3} \text{ \AA}$ yields an upper limit of 4.4 nm).

If the authors address these critical issues, the manuscript may be suited for Nature.

Referee #3 (Remarks to the Author):

In this manuscript, the authors aimed to study the molecular mechanisms underpinning the action of MOR ligands. To better understand the specific conformational changes induced by a variety of opioid ligands, they combined DEER and single-molecule FRET technologies and found that receptor conformations are interconverting. More specifically, they demonstrate that coupling efficacy to downstream effectors (G protein and b-arrestin) is determined by the ligand-dependent conformational dynamics of the receptor.

The study is original and novel in the fact that DEER experiments revealed that the co-existing conformations of MOR can be modulated by ligands with different efficacy. Most interestingly, these experiments revealed a discrepancy between the canonical active receptor population and the ligand efficacy, suggesting that the TM6 movement is not sufficient to define the activity of the receptor.

The study is important in that it sheds light on receptor conformation dynamics and how ligand can interfere and modulate the conformational scheme. Additionally, the study revealed unprecedented molecular mechanisms responsible for ligand-dependent coupling efficacy to downstream effectors.

Comments and concerns:

In order to use DEER and single-molecule FRET, the authors have modified the receptor. They mention (page 3; lines 4-5) that cysteine mutations did not alter the agonist and antagonist binding properties on MOR. However, only displacement curves with DAMGO and diprenorphine are shown in Figure S4. The entire study relies on "small" conformational changes. Cysteine mutations may produce small changes that could potentially impact the binding affinity of a ligand more than another. For this reason, measuring the affinity of all ligands for the mutant receptors would strengthen the study.

DAMGO is known to bind MOR with nM affinity. It is therefore surprising (cf. Fig S4) that the authors are reporting affinity for DAMGO in the μM range, even for the WT-MOR. This apparent discrepancy with the literature should be explained, in particular with respect to Fig. 1C-E where DAMGO activates G_i with an efficacy in the range of 10 nM.

Another thing that is missing is a demonstration that the mutant receptors are functionally coupled to G_i and b-arrestin. This is particularly true for b-arrestin as the experiments cannot confirm that it binds to the mutant receptors.

Minor comment:

It is unclear from Fig. S4 and the associated method section if the authors calculated the K_i value. They report on the graph values as being K_i (and pK_i), however, given the high concentration of $[^3\text{H}]$ -diprenorphine in the experiments, K_i does not equal IC_{50} . Please specify if Cheng-Prusoff equation had been applied.

Louis Gendron

Author Rebuttals to Initial Comments:

Referee #1: GPCRs, including single molecule methods

Review of

Jiawei Zhao, Matthias Elgeti et al.

Conformational dynamics of the μ -opioid receptor determine ligand intrinsic efficacy

The teams of Chunlai Chen, Brian Kobilka and Wayne Hubbell have combined efforts in order to delineate the dynamics of a G-protein-coupled receptor (GPCR). Using two highly sensitive methods, double electron-electron resonance (DEER) and single-molecule fluorescence resonance energy transfer (smFRET), the authors explore conformations of the μ -opioid receptor - a prototypical class A GPCR that mediates most of the wanted as well as unwanted effects of opioids. This receptor is interesting not only because it is of great clinical importance and has been studied in quite some detail, but also - in the context of the present work - because ligands with very different properties (different chemical nature, partial vs. full agonism, biased activation etc.)

The background of this work is formed by published X-ray and cryo-EM structures of the receptor alone or with its cognate G protein, Gi. These structures, however, most likely represent only stable states that may or may not reflect the most relevant conformations. In addition, many more conformations may exist that are not stable enough to be amenable to crystallization or cryo-EM.

Although there are variations in details between the constructs used in the different experiments, the authors concentrate on a single axis in the receptor, i.e. TM4 – TM6. While this is logical based on the fact that the outward movement of TM6 is a hallmark of GPCR activation, it means that conformational heterogeneity is limited to a 2-dimensional view.

DEER studies could be resolved into 6 different peaks, of which 4 were deemed functionally relevant. Ligands produced only minor changes in these populations, with the exception of the super-agonists lofentanil and BU72, which led to marked increases in the long distance populations, compatible with the outward movement in TM6 during activation. In the additional presence of Gi, such an increase in the long distance populations became visible for all other ligands (with the obvious exception of the antagonist naloxone), suggesting that Gi stabilized an active receptor conformation. Interestingly, this active conformation was characterized for most ligands with roughly equal proportions of the 39Å and the 43Å populations (suggesting that the active G-protein-coupled state is composed of a mixture of these two states), while lofentanil and BU72 alone had led to a clear preponderance of the 43Å population. Also of interest is the observation that Gi alone had little effect in these experiments, indicating that while it facilitated formation of an active state, it did not do so on its own.

In contrast to the major effects of Gi on these different receptor populations, the effects of beta-arrestin1 were more modest (compared to the respective ligands alone). This suggests that the effects of beta-arrestins on receptor conformations are much less pronounced than those of the G-protein(s).

smFRET experiments were then conducted to study the kinetics of interconversion between different receptor populations. Unfortunately, this method is also limited in temporal resolution, in this case to 100 ms, which leads to averaging between rapidly (i.e. <100ms) exchanging conformations. As a consequence, experiments with Cy3/Cy5 labeled receptors revealed only a shift in mean FRET that was observed with DAMGO and BU72 (low FRET indicating again outward movement of TM6) - roughly in line with the DEER data for superagonists and suggesting the presence of rapidly achieved active state(s). With the Cy3/Cy7 construct, an (additional) slower (i.e. >100ms) conformational change could be resolved. Such fast vs. slow conformational changes would be compatible with receptor activation kinetics observed in intact cells, where receptor activation itself occurs with time constants <100ms, whereas G-protein-dependent effects occur at time scales >100ms.

We thank Reviewer 1 for his/her positive comments. While we generally agree with his/her summary of our findings, we would like to emphasize that the slow conformational exchange (>100ms) was observed using single-molecule FRET of μ OR-Cy3/Cy7 as shown in Figure 3C in the absence of G protein and thus also reflects “receptor activation itself”. This is notable as it suggests that the receptor may control slow, potentially rate-limiting steps for G protein dependent signaling.

In page 7 line 40 of the discussion section, we modified the following sentence to emphasize this point “Our data revealed a slow conformational change with an exchange rate dwell time > 100 ms connected to receptor pre-activation, a structural change distinguishing μ OR bound to the antagonist naloxone and low-efficacy G protein biased agonists, which is a potentially rate-limiting step for G protein and β -arrestin binding and signaling.”

smFRET experiments at different concentrations of GDP led the authors to two main kinetic conclusions: (a) super-agonists increase the speed of binding of G_i to the receptor, and (b) G-protein-biased ligands do not lower GDP-affinity for G_i . Finding (a) is compatible with data from receptor/ G_i -interactions in intact cells, where agonists have been reported to increase the k_{on} of binding. I find it hard to understand finding (b), because I would assume that even G-protein-biased ligands would need to induce GDP-release from G_i ; how would this happen if the affinity is not lowered?

We completely agree, binding of active μ OR also lowers the GDP affinity for G_i when bound to low-efficacy G protein biased agonists. However, when bound to these ligands the effect is much smaller as compared to super-efficacy agonists. We rephrased the following sections of the results and discussion:

Page 6, line 45: “Taken together, G protein-biased agonists fail to lower GDP affinity to G_i as dramatically as high- and super-efficacy agonists, which, in combination with slower G_i binding (Figure 4F), manifests in their lower efficacy.”

Page 8, line 3: “Low-efficacy, G protein-biased agonists lead to a slower release of GDP and large fractions of the complex remain GDP-bound.”

As in DEER experiments, no major specific effects of beta-arrestin1 were observed, confirming the notion of a less tight (and perhaps specific) nature of this interaction. It might be interesting to assess in these experiments, whether the same would be true for beta-arrestin2, which in several instances has been reported to bind receptors more tightly.

This is a valuable suggestion, however, we believe that including experiments in the presence of beta-arrestin-2 is not necessary. First, in our DEER experiments, 100 μ M or higher of beta-arrestin-1 is used, which is significantly higher than the saturation concentrations used for all ligands we tested (as shown in our Figure 1D). Second, beta-arrestin-1 did induce conformational changes in the μ OR. Although the beta-arrestin-1-induced changes are relatively small, they are still statistically significant differences between the distance distributions with and without beta-arrestin-1 (Figure 2). Finally, in our BRET-based assays, beta-arrestin-2 exhibits similar behavior as beta-arrestin-1 (Figure R1).

Thus, due to the similarity in sequence and function, we do not expect large structural differences in receptor structure upon coupling to beta-arrestin-2. We agree that the minor changes induced by beta-arrestin could be due to the binding promiscuity of arrestins which are likely to have a weaker interaction with any given GPCR.

Figure R1. Intrinsic efficacy of ligands towards β -arrestin-2 and β -arrestin-1 determined by BRET-based assays

Overall, this is a very well conducted, carefully performed and interpreted study by presumably the best team of authors that might address these issues. Aside from the considerations above I have no technical criticisms. I think, three aspects might be improved in the presentation/discussion:

1. For many aspects there is more data in the literature to support the authors' conclusions; a more encompassing discussion of this literature might help the authors to make their points, and might help the reader to position the present manuscript; this refers to both, experiments with isolated reconstituted systems including smFRET and experiments in intact cells;

2. It would be good if the authors might dare to move a little further than just interpret their data in terms of TM4-TM6 distances; what do we learn about the 3D cytosolic interface of the receptor in these different states?
3. It would be helpful if the authors could go a little further in explaining what they learn from their experiments beyond what is known from earlier NMR experiments, which have shown already the dynamic nature of inactive and active receptor conformations, the presence of multiple conformations that contrast with simple on/off-models etc.

Thank you for your suggestion.

We included the following sentences in the result section to briefly summarize related smFRET studies in the literature.

Page 4, line 24 “...we performed smFRET experiments, which have been utilized to capture the conformational dynamics of the $\beta_2AR^{29,30}$, the metabotropic glutamate receptor dimer³¹, and β -arrestin^{32,33} using reconstituted systems or in cell membranes. Using an experimental design similar to that previously reported for the β_2AR^{29} , smFRET of labeled μOR , despite the lower spatial resolution compared to DEER, provides access to protein dynamics...”

In the conclusion section, we added the following sentences to summarize recent findings in recent NMR, MD and DEER studies and emphasize our new insights.

Page 7, line 15 “Studies using nuclear magnetic resonance (NMR) spectroscopy, molecular dynamics (MD) simulations, and DEER indicate that the conformational dynamics of GPCRs, especially in the TM5, TM6, TM7, ICL1, ICL2 and H8 domains^{39,46-48}, play important roles in GPCR functional selectivity. Our results reveal the TM6 conformational heterogeneity and both fast and slow conformational dynamics of TM6 and ICL2 are differentially modulated by distinct ligands.”

Referee #2 EPR/DEER:

A: The authors report in their manuscript on a DEER smFRET study on the conformational states of the μ -opioid receptor upon binding of several different pharmaceutically relevant ligands in the presence and absence of two transducers, i.e., G protein G_i and β -arrestin-1). Their data are supported by ligand binding essays. They identify several new states and show that these interconvert on largely different time scales. They identify the molecular basis of differences in the action of the ligands and the influence of the ligands on the transducer action.

B: The combination of DEER and smFRET is highly interesting but not novel (see paper from Hagelueken et al. Nat Commun 2022). The application to the μ -opioid receptor is new, and the system as such highly relevant. The insight gained here is in my opinion a quantum leap in the understanding of this receptor and for the action of pharmaceutica in general. The found molecular principles

We thank the reviewer for his/her positive feedback and we have added the referenced publication in the introduction (Page 2, Line 29).

C: The combination of DEER as an ensemble technique that yields the distribution of the whole ensemble and smFRET providing time scales and real time movements is highly appropriate for the question asked. Indeed, only the combination of the three methods (DEER, smFRET, and biochemical assays) made it possible to draw the conclusions that "TM6 movement alone does not define receptor activity". The data and their presentation is of high quality.

D: The data statistics is valid as performed, but:

1) The label HO-1427 is restricted in mobility, one should therefore expect orientation selective DEER time traces. Using a chirp pump pulse will reduce orientation selection but the detection pulses are still selective. Since some of the observed changes in the distance distributions are very subtle, it would need to be made sure that in none of the cases the onset or diminishing of orientation selection occurs e.g., by recording DEER time traces at different offsets from the pump pulse. If orientation selection is present this needs to be considered e.g., by summing up time traces at different offsets or by taking it explicitly into account in the analysis. This is crucial for the whole analysis.

We understand the reviewer's scrutiny towards the new HO-1427 spin label. However, we took great care choosing a suitable spin label, as described briefly in the following, and thus exclude the possibility of orientation selection in our dataset:

Initial tests of spin labeling and DEER on μ OR were performed using the methanethiosulfonate (MTSSL) or iodoacetamide proxyl (IAP) spin labels (see Figure R2), which represent the established spin labels for DEER analysis of GPCRs devoid of orientation selection ([10.1073/pnas.1620405114](https://doi.org/10.1073/pnas.1620405114) or [10.1016/j.cell.2015.04.043](https://doi.org/10.1016/j.cell.2015.04.043)). The modulation depth for both spin labels was about 40-45% for both labels and all ligands tested, which is a value we commonly observe on our experimental setup. Since MTSSL was cleaved in the presence of G protein and IAP exists as a racemic mixture (leading to broader distance peaks), we tested HO-1427 as an alternative label. HO-1427 combines the positive characteristics of MTSSL and IAP, such as high

spatial resolution and a non-cleavable thioether linkage. We again observed an almost identical modulation depth for HO-1427 under inactive and active receptor conditions (see Figure R2). If orientation selection was present in our dataset, one would expect that the modulation depths would differ for the different labels and different conformations, which was not the case. We are therefore confident that our distance distributions and ligand induced changes are real and not skewed by orientation selection. We want to emphasize that orientation selection is a phenomenon which occurs only when either the label linker or the label environment drastically reduces the conformational space of the spin label so that motional averaging of the dipolar vectors cannot occur. This becomes apparent in DEER signals, which cannot be fitted by regular fitting algorithms, leading to a poor fit, especially in the time window $<1\mu\text{s}$. Since GPCRs are highly dynamic proteins we do not expect that even the most rigid spin labels would lead to detectable orientation selection.

Figure R2. Comparison of different spin labels for DEER analysis of mOR-180C-276C. Colors as indicated in the legend. DEER traces, model-free analysis fits and distributions were shifted for easier comparison. Please note that the protein construct differs slightly from the one reported in the manuscript (mOR-182C-276C).

In order to further investigate potential influence of orientation selectivity in the presented HO-1427 labeled samples, we also performed the suggested DEER experiment using various different frequency offsets (Figure R3). We chose Lof+Gi for these exemplary experiments as our DEER data suggests that mOR is most rigid in this state due to the stabilization via ligand and G protein, and if orientation selectivity was present it would be most apparent under these conditions. Our data suggests a stable modulation depth of $\sim 40\%$ and consistent distance distributions for all frequency offsets, which provides further evidence for the validity of the chosen analysis method and interpretation.

Figure R3. DEER results for HO-1427 labeled mOR bound to Lofentanil/Gi using several different offsets between observer and pump frequencies. Consistent modulation depths and distance distributions across the different frequency offsets excludes the occurrence of significant orientation selectivity distorting the observed DEER signals.

2) The authors do not present repeats but argue that the whole data as such is consistent and represents as such the repeats. I do not agree with this argument because it is circular. Their analysis relies on the whole data set being correct. Repeats would also give a better idea about

the error of the experiment. Also, according to the white paper of the community (J. Am. Chem. Soc. 2021, 143, 17875–17890) repeats must be presented (mixture of technical and biological ones).

Figure R4 (new Figure S13D). Biological repeats for two selected ligand and transducer conditions. No major differences are observed.

Biological repeats: As described for comment 1, we put a great deal of work into the selection of μ OR construct, spin label, ligand/transducer conditions and data analysis before recording the final, reported DEER dataset. Since μ OR expresses in very low amounts (~40ug/L from ~3L/sample) we had to limit the number of samples for economic and practical reasons.

Nevertheless, we performed biological repeats for Naloxone and Lofentanil with and without Gi, which represent the most distinct ligand/transducer conditions (Figure R4, new Figure S13D). Overall, we obtain very good

reproducibility of the model-free distance distributions. In particular, for both ligands, the smaller G_i -induced shift are reproduced accurately.

Technical repeats: All DEER samples were prepared in parallel, each step by the same experimenter and using the same protocol. The samples were aliquots of the same batch of receptor and frozen immediately after the final concentration step.

Since μ OR and GPCRs in general exhibit low stability and tend to aggregate upon thawing and refreezing of the sample, especially when prepared in detergent solutions, we did not perform technical repeats on freezing.

All DEER experiments were set up by the same experimenter using identical pulse sequences and parameters. We show here the results of two independently set-up experiments (the sample was taken out of the instrument, stored on LN₂, and rerun a few days later). The results are highly correlated ($R=0.9959$, Figure R5) highlighting the reliability of our setup.

We think that the presented combination of biological and technical repeats are sufficient to verify the presented DEER dataset.

3) Connected with 2), the authors find discrepancies between DEER and smFRET for DAMGO and G protein-biased agonists.

The argument that a long flexible linker should increase a small structural change eludes me. Commonly, a small and rigid label can report on a small structural change but not a label that is connected via a long flexible linker to the site of the structural change. This is a major inconsistency. If the reason is the different label site, then this should be checked by labeling the same sites.

Figure R5. Technical repeat of buprenorphine sample.

As the reviewer correctly notes, the smaller EPR labels provide higher spatial resolution in capturing translational motions than the larger Cy-labels used for the smFRET experiments. However, when rotational motions of a helix occur as shown in the following scheme (Figure R6), a longer linker may indeed amplify a conformational change. During canonical GPCR activation, TM6 undergoes both translational and rotational motions (as shown in Figure S16), of which the latter is likely to be amplified by the longer linker.

4) Connected with 2) and 3), the whole analysis is based on one label pair, this is not enough but must be complemented by additional pairs at least one better two.

We present results for two fluorophores and labeling sites (Cy3/Cy5 labeled at 182C/273C and Cy3/Cy7 labeled at 180C/276C), for which we were able to record smFRET data with sufficient signal quality. During our initial test of smFRET experiments, 4 labeling constructs and 12 FRET dye pairs were prepared and screened. Although their results were

Figure R6. Schematic of amplified distance change due to a longer label linker.

not presented in the manuscript, two constructs shown in Figures 3 and 4 give the clearest FRET changes upon the addition of different ligands. For instance, when $\mu\text{OR-Cy3/Cy5}(180\text{C}/276\text{C})$ was used to replace $\mu\text{OR-Cy3/Cy7}(180\text{C}/276\text{C})$, we still captured changes of FRET in the presence of different ligands and upon binding of the G protein (Figure R7). However, the changes of FRET efficiencies of $\mu\text{OR-Cy3/Cy5}(180\text{C}/276\text{C})$ in the different ligands were smaller than those of $\mu\text{OR-Cy3/Cy7}(180\text{C}/276\text{C})$. Thus while Cy3-Cy5 is sensitive to longer distances and presumably resolved TM6 outward tilt, Cy3-Cy7 exhibits a shorter Forster radius and was thus able to resolve changes of TM6 and ICL2 that occur on a different length- and timescale. Together, we would like to emphasize that the fluorophores and FRET pairs used in smFRET exhibit considerable linker lengths, conformational flexibility and distinctive Forster radii, and are thus likely to explore and report on different motions of the intracellular receptor surface.

Figure R7. FRET distributions of $\mu\text{OR}\Delta 7$ -180C/276C labeled by Cy3/Cy7 (A) or Cy3/Cy5 (B). (A) is from Figure 3C in the manuscript.

5) The control experiment for checking whether the remaining cysteines in the minimal cysteine mutant are labeled was done with Iodoacetamido proxyl while the labeling itself was then done with HO-1427. This control experiment must be repeated with the actual label.

We repeated the labeling experiment of the minimal cysteine construct $\mu\text{OR}\Delta 7$ using HO-1427. We did not find any significant labeling (<5%). We updated current Figure S5 accordingly.

E: At the current stage the missing data leave for me a large question mark on the robustness, validity and reliability of the interpretation of the data and conclusions.

We are in full and enthusiastic support of the recent developments towards reproducible science. Therefore, even though there is currently no standard in the field, we present a combination of biological and technical repeats of our DEER experiments. The biological repeats reliably reproduce even the small Gi-induced distance changes for naloxone and lofentanil. Unfortunately, due to economic reasons we had to limit the number of samples recorded for the exact same construct, spin label and conditions. For our single-molecule FRET measurements, as mentioned above, 24 different combination of protein constructs and labeling sites were tested. Among them, two of them provide the clearest FRET change and best signal quality are chosen for further examination as shown in Figures 3 and 4 of our manuscript.

Overall, both our DEER and single-molecule FRET measurements provide robust and reliable results and conclusions, which can be validated by different constructs (with FRET pairs as shown in Figure R7), and necessary biological and technical repeats (as shown in Figures R4 and R5).

F: Fig. S11 please use the actual label (HO-1427) in mtsslWizard (the author of the program and host of the site is always willing to include new labels on request if he hasn't done so already).

We updated the figure and HO-1427 is now included in the mtsslWizard.

G: 1) Give references for DEER and smFRET. I would suggest taking the two white papers of the respective communities. 2) Give a reference for mtsslWizard.

We added the respective citations in the introduction.

H: Clarity and context are fine. Two minor things: 1) Fig. S10. Caption: I suspect the authors mean $p < 0.5$ instead of 0.05.

The figure and caption are correct. We highlighted statistically significant correlations with a p -value < 0.05 .

2) The authors state that they did not include distances above 4.4 nm into the analysis, which is good. But when discussing the reasons please also include a statement that the lengths of the time traces exclude their analysis ($r_{max} = 30 (t_{max}/\mu s)^{1/3}$ Å yields an upper limit of 4.4 nm).

The referenced criterion for r_{max} is valid only for non-parametric analysis of single time-traces. Global analysis of parametric fits significantly can extend the confidence window, however, a clear cutoff is difficult to define as it depends on many factors of the dataset to be analyzed (number of traces, signal to noise, number of conditions, differences between conditions etc.). We therefore performed an error analysis using bootstrapping which provides confidence bands for all fit parameters and distance distributions (<https://doi.org/10.5194/mr-1-209-2020>).

If the authors address these critical issues, the manuscript may be suited for Nature.

Referee #3: opioid receptor pharmacology

In this manuscript, the authors aimed to study the molecular mechanisms underpinning the action of MOR ligands. To better understand the specific conformational changes induced by a variety of opioid ligands, they combined DEER and single-molecule FRET technologies and found that receptor conformations are interconverting. More specifically, they demonstrate that coupling efficacy to downstream effectors (G protein and b-arrestin) is determined by the ligand-dependent conformational dynamics of the receptor.

The study is original and novel in the fact that DEER experiments revealed that the co-existing conformations of MOR can be modulated by ligands with different efficacy. Most interestingly, these experiments revealed a discrepancy between the canonical active receptor population and the ligand efficacy, suggesting that the TM6 movement is not sufficient to define the activity of the receptor.

The study is important in that it sheds light on receptor conformation dynamics and how ligand can interfere and modulate the conformational scheme. Additionally, the study revealed unprecedented molecular mechanisms responsible for ligand-dependent coupling efficacy to downstream effectors.

We thank the reviewer for his/her time, effort and the valuable comments to our manuscript. We are glad he/she understands the impact of our work.

Comments and concerns:

1) In order to use DEER and single-molecule FRET, the authors have modified the receptor. They mention (page 3; lines 4-5) that cysteine mutations did not alter the agonist and antagonist binding properties on MOR. However, only displacement curves with DAMGO and diprenorphine are shown in Figure S4. The entire study relies on "small" conformational changes. Cysteine mutations may produce small changes that could potentially impact the binding affinity of a ligand more than another. For this reason, measuring the affinity of all ligands for the mutant receptors would strengthen the study.

We agree that receptor function should be verified for the cysteine constructs. This is often done by determining ligand binding using saturation binding or competition ligand binding, as presented in Fig. S4 for exemplary ligands (antagonist and agonist). Another way to test receptor function is to investigate ligand-dependent G protein binding, as presented in Fig. 2B (DEER, with large excess of ligand) and Fig. 4 (smFRET). Moreover, we find that our smFRET results of Cy3/Cy7 labeled μ OR correlate very well with ligand efficacy towards G protein activation of wildtype (Fig. 1C) and labeled μ OR (Fig. 4C-G).

It is noteworthy that four of the native cysteines which were removed in μ OR Δ 7 are located in the disordered N terminus, and the remaining three cysteine mutations (Cys170, Cys351, and Cys346) are on the intracellular side (Figure R8). None of these residues are in contact with ligands or transducers, thus we are confident that differences in ligand affinity are negligible.

Figure R8 (current Figure S2). Snake plot of the μ OR sequence. Cysteine residues in yellow indicate native cysteine that were kept in the labeling constructs. Residues in green indicate native cysteine that are mutated to corresponding residues to avoid nonspecific labeling by cysteine reactive labeling reagent. Residues in red indicate labeling sites that are mutated to cysteines for labeling with cysteine-reactive fluorophores and nitroxide spin labels.

We further performed the competition binding assay to evaluate the function of the minimal-cysteine μ OR (Figure R9, new Figure S4), indicating the IC_{50} of each ligand is mildly affected by cysteine mutations. Similar effects were found when examining minimal-cysteine AT1R (Wingler, 2019, Cell)

Figure R9, new Figure S4. Competition binding of wild-type ($\mu\text{OR-WT}$) and minimal-cysteine ($\mu\text{OR}\Delta 7$) μOR . The cell membrane of *sf9* cells expressing $\mu\text{OR-WT}$ or $\mu\text{OR}\Delta 7$ was extracted and used for competition binding. The hot ligand is $^3\text{H-naloxone}$.

Moreover, since our study is aimed to characterize the molecular background of ligand efficacy, all our DEER and smFRET experiments were performed at saturating ligand concentrations.

Finally, the best proof of receptor function is presented by our GDP titrations using smFRET (Figures 4C-G), which show the entire signaling module of ligand, μOR and G protein “at work”. The results of G protein binding and GDP release fit into established structure/function relationships of GPCR mediated signaling.

2) DAMGO is known to bind MOR with nM affinity. It is therefore surprising (cf. Fig S4) that the authors are reporting affinity for DAMGO in the μM range, even for the WT-MOR. This

apparent discrepancy with the literature should be explained, in particular with respect to Fig. 1C-E where DAMGO activate Gi with an efficacy in the range of 10 nM.

In this work we reported the K_i of DAMGO to the μ OR and its mutants in sf9 cell membrane at 3.3–17 μ M in the presence of 2.9 nM 3 H-DPN. The reported DAMGO K_i values are orders of magnitude higher than its potency determined by TRUPATH assay. There are a few reasons:

a) *Stahl et al (DOI: 10.1073/pnas.2102178118, Table 3) showed that K_i of cold ligands in the presence of different hot ligands of 3 H-DAMGO and 3 H-DPN are at least one order of magnitude different and larger than their potencies. Therefore, the value of DAMGO K_i reported by the competition assay might be higher than its potency determined by TRUPATH assay.*

b) *In our radioligand binding assays, we are using insect cells (sf9) membrane, which do not have intracellular transducers that can couple to the μ OR. By contrast, most cell functional assays use HEK293 cells, which express G protein that can couple to the μ OR, hence, allosterically enhances the agonist binding affinity. This allosteric effect has been demonstrated by DeVree et al (DOI: 10.1038/nature18324). Consistently, Massotte et al (DOI: 10.1074/jbc.272.32.19987, Table IV) reported K_i of DAMGO to sf9-expressed human μ OR as 378 nM in the presence of 3 H-DPN, which stands in contrast to 0.9 nM in COS cells (DOI: 10.1523/JNEUROSCI.15-03-02396.1995), suggesting cell lines may significantly affect the measured K_i of agonists.*

3) *Another thing that is missing is a demonstration that the mutant receptors are functionally coupled to Gi and b-arrestin. This is particularly true for b-arrestin as the experiments cannot confirm that it binds to the mutant receptors.*

The presented DEER data suggests that the labeled receptor constructs couple efficiently to G protein, which quantitatively stabilizes the active receptor conformation when coupled to agonists (Fig. 2B). Fig. 2C shows that β -arrestin1 can change the distance distributions of spin-labeled μ OR significantly, suggesting that β -arrestin-1 is interacting with the μ OR. However, compared to G proteins, β -arrestins have intrinsically low affinity for core interaction with μ OR, and induces smaller conformational changes of μ OR than G protein even with saturation concentration. This is why it is very challenging to prepare a stable in vitro complex and hence the μ OR/ β -arrestin-1 complex has eluded structure determination.

We agree, it is likely that only a small fraction of β -arrestin-1 is engaged in the core interaction with μ OR, and thus stabilizes the active conformation in DEER. Another possibility is that, as in our response to the Reviewer #1, the minor changes induced by beta-arrestin could be due to the binding promiscuity of arrestins which are likely to interact with several receptor conformations and thus have less effect on the conformational equilibrium of the receptor. Nevertheless, we were still able to resolve small but significant differences in the DEER binding pattern using saturating β -arrestin-1 compared to G protein. We thus included the β -arrestin-1 data in this manuscript, as it constitutes incremental insight into the diverse interaction patterns between active GPCRs and their transducers.

We further performed the BRET assay to evaluate the function of the minimal-cysteine μ OR (Figure R10, new Figure S3). The EC_{50} and efficacy of each ligand slightly changed compared to the wild-type μ OR, but none ligand has changed their category in terms of ligand bias or efficacy, which suggests that the μ OR mutant can reflect the wild-type μ OR functions in this study scope.

Figure R10 (new Figure S3). Functions of wild-type ($\mu\text{OR-WT}$) and minimal-cysteine ($\mu\text{OR}\Delta 7$) μOR determined by TRUPATH assays.

4) Minor comment:

It is unclear from Fig. S4 and the associated method section if the authors calculated the K_i value. They report on the graph values as being K_i (and pK_i), however, given the high concentration of [3H]-diprenorphine in the experiments, K_i does not equal IC_{50} . Please specify if Cheng-Prusoff equation had been applied.

More details of the method for radioligand binding assays now are added to the Method section. We used Prism (GraphPad) to analyze the radioligand binding data. The competition binding data were fitted in a one-site (fit K_i) model in Prism. According to GraphPad (https://www.graphpad.com/guides/prism/latest/curve-fitting/reg_one_site_competition_ki.htm), the model is:

$$\log\text{EC}_{50} = \log(10^{\log K_i} (1 + \text{RadioligandNM} / \text{HotKdNM}))$$

$$Y = \text{Bottom} + (\text{Top} - \text{Bottom}) / (1 + 10^{(X - \log\text{EC}_{50})})$$

In which, Top and Bottom are plateaus in the units of Y axis, RadioligandNM is the concentration of labeled ligand in nM, and HotKdNM is the equilibrium dissociation constant of the labeled ligand in nM. According to GraphPad, "This model fits the K_i of the unlabeled ligand directly. It does not report the EC_{50} , so you do not need to apply the Cheng and Prusoff correction. Instead you enter the concentration of radioligand and its Kd as constants, and Prism directly fits the K_i of your cold compound.' Therefore, we directly calculated K_i .

Reviewer Reports on the First Revision:

Referees' comments:

Referee #1 (Remarks to the Author):

I am happy with the arguments in the authors' rebuttal letter and with the changes they made in the manuscript in response to the criticisms and suggestions raised by all 3 reviewers. In particular, I accept their argument that studies with beta-arrestin-2 would presumably not lead to new conclusions.

The only point where I would like to invite the authors (again) to add sth to their manuscript would be a more general conclusion at the end of the manuscript. I feel that the statement "the present study provides novel insights..." is below what the authors have observed. It would be nice to state with a few sentences at the end of the text what the less specialist reader should learn from this interesting paper.

Referee #2 (Remarks to the Author):

The authors addressed all of the points raised by all three reviewers and I agree with all of their arguments and new experiments apart from two:

Reviewer #2, point 3: I don't agree with the authors that a longer label resolves a small change better than a short one even if the change is due to a rotation. My point is that a longer, flexible linker, as used here, induces more conformational flexibility on top of the small change that is to be resolved. The linker is not a rigid stick, but comprised of several rotatable bonds, leading to a large conformational space. This together with the issue that two different sites for the labels were used leaves a major inconsistency that must be addressed. The only way that I see is to use the same label sites and to resolve the change with both methods. As it stands, the observation that the small label does not resolve the change whereas the flexible linker provides a change may rather point to changes in the conformer space of this label than the protein.

Reviewer #2, point 4: This is related to point 3 above. Also for DEER two label pairs have to resolve the conformational change. I understand that the production of the protein is difficult but that is why I am fine with 2 sites for FRET but these sites have to be corroborate with two sites for DEER, especially if there is a discrepancy between the methods for one pair. I am sorry that I was a bit unclear on this point in the previous review.

Reviewer #2, E: The authors are to be commended for their repeats and the large amount of work they had to put into it. I am fully satisfied with regard to this point. Having said that and without implication for this manuscript: I disagree with their reply that there is no standard in the field. The white paper of the DEER community does clearly state that repeats (in form of a combination of technical and biological repeats) are a must.

Reviewer 2, H, point 2: I see, well done.

Olav Schiemann

Referee #3 (Remarks to the Author):

I thank the authors for their responses. The additions made to the manuscript, together with their point-by-point response, are much appreciated. I am satisfied with the corrections.

Author Rebuttals to First Revision:

Referee #1 (Remarks to the Author):

I am happy with the arguments in the authors' rebuttal letter and with the changes they made in the manuscript in response to the criticisms and suggestions raised by all 3 reviewers. In particular, I accept their argument that studies with beta-arrestin-2 would presumably not lead to new conclusions.

The only point where I would like to invite the authors (again) to add sth to their manuscript would be a more general conclusion at the end of the manuscript. I feel that the statement "the present study provides novel insights..." is below what the authors have observed. It would be nice to state with a few sentences at the end of the text what the less specialist reader should learn from this interesting paper.

We thank reviewer #1 for the time and effort spent on the review. Toward his/her comment about a more general conclusion we added the following sentences to the final paragraph of the manuscript:

' ...as we report the first experimental evidence for important intermediate conformations, responsible for G protein functional selectivity. These findings point towards exciting new avenues for the design of therapeutics with fewer adverse effects, targeting sparsely populated conformational states which so far have escaped high resolution structural biology methods. While the need for these therapeutics is imminent for the opioid receptor subfamily, intermediate conformations with functional selectivity properties have been reported for other GPCRs ([10.1016/j.cell.2018.12.005](https://doi.org/10.1016/j.cell.2018.12.005)) and thus may represent a general approach to drug design.'

Referee #2 (Remarks to the Author):

The authors addressed all of the points raised by all three reviewers and I agree with all of their arguments and new experiments apart from two:

Reviewer #2, point 3: I don't agree with the authors that a longer label resolves a small change better than a short one even if the change is due to a rotation. My point is that a longer, flexible linker, as used here, induces more conformational flexibility on top of the small change that is to be resolved. The linker is not a rigid stick, but comprised of several rotatable bonds, leading to a large conformational space. This together with the issue that two different sites for the labels were used leaves a major inconsistency that must be addressed. The only way that I see is to use the same label sites and to resolve the change with both methods. As it stands, the observation that the small label does not resolve the change whereas the flexible linker provides a change may rather point to changes in the conformer space of this label than the protein.

- 1) Linker length and flexibility should be considered separately: While a flexible linker lowers resolution, a longer linker may amplify conformational changes involving rotations of protein segments (cf. figure in last response). Moreover, a long linker does not necessarily make the fluorophore more flexible. The same fluorophore linker was used in β_2 AR in the

previous study (DOI: 10.1038/nature22354), and the distance between the fluorophores and that between the C α carbons of the labeling sites were investigated using MD simulations (Extended Data Figure 5a and b of DOI: 10.1038/nature22354). According to their MD simulations results, even with the same long fluorophore linkers as in our study, the change of the distance between two fluorophores caused by TM6 movement is almost the same as the distance change between C α carbons at the labelling site. This is probably because, although the linker is long and there are potentially many rotatable bonds, the actual flexibility of the linker is restricted by its environment. As a result, the linker does not amplify the actual conformational space. Thus, the influence of the long flexible linker may vary case by case.

- 2) As the reviewer suggested, we performed additional smFRET measurements with the same labeling sites as those used in the DEER experiments (R182/R276) (Figure R1; these new data are added to Extended Data Fig. 15 in the manuscript now). The smFRET results of the μ OR labeled at R182/R276 with Cy3/Cy5 or Cy3/Cy7 are similar to the DEER results showing very small changes in distance for partial agonists compared to full agonists (Figure R1 D-F). Cy3/Cy7- μ OR Δ 7-R182/R276 showed larger FRET changes for different ligands than Cy3/Cy5- μ OR Δ 7-R182/R276, emphasizing the importance of choosing the FRET pair with optimal Förster radius (R0 values in current Extended Data Fig. 8f) to capture subtle but important conformational changes. In addition, the two FRET states shown in Cy3/Cy7- μ OR Δ 7-R180/R276 in the presence of low-efficacy ligands (Fig. 3c) cannot be resolved in Cy3/Cy7- μ OR Δ 7-R182/R276 (Fig. R1D, current Extended Data Fig. 15f), supporting our assignment that these two FRET states are caused by slow structural transition in ICL2. Moving one labeling sites from T180 to R182 (away from ICL2) decreases its sensitivity towards local conformational changes in ICL2.

Figure R1. smFRET results using the DEER construct ($\mu\text{OR}\Delta 7\text{-R182/R276}$). $\mu\text{OR}\Delta 7\text{-R182/R276}$ is labeled with Cy3/Cy5 (A-C) and Cy3/Cy7 (D-F), respectively. (A) and (D), FRET distributions. (B) and (E), FRET peak centers in (A) and (D), respectively. Numbers on top of each bar are the peak centers extracted from the Gaussian fitting. Error bars indicate standard errors of the fitting. (C) and (F), averaged FRET values in (A) and (D), respectively. FRET efficiencies between 0.6 and 1.2 in (A) and between 0 and 1.2 in (D) were used for calculation. Error bars indicate s.e.m. *, $p < 0.001$.

In summary, long linkers of fluorophores can still faithfully report the change of relative distance between FRET labeling sites (DOI: 10.1038/nature22354), and therefore do not hinder the detection of small conformational changes. In fact, as discussed in our previous response letter, the long linker may amplify the rotational conformational changes compared to a short linker. In general, the FRET technique is less accurate in measuring distances between two labeling sites than DEER. However, as suggested by the reviewer, the accessible volume (conformational space) of the fluorophores with long-flexible linkers may be more sensitive to the local environment of the fluorophores, caused by nearby local or global conformational changes. Thus, sensitivity to rotation and/or local environment, likely contribute to the differences observed for the different labeling sites, fluorophores, and methods. DEER and smFRET are complementary methods in detecting the conformational complexity of the μOR upon binding to different ligands and transducers.

We modified the following sentences in page5, starting in line33:

'Cy3/Cy5 and Cy3/Cy7 labeled μ OR Δ 7-R182C/R276C, the same construct used in our DEER measurements (Extended Data Fig. 15c-h), displayed the similar trend of FRET changes in the presence of a series of ligands. However, μ OR Δ 7-R182C/R276C-Cy3/Cy7 is unable to resolve two FRET states shown in μ OR Δ 7-R180/R276-Cy3/Cy7 in the presence of low-efficacy ligands (Fig. 3c). This finding supports our assignment that these two FRET states reflect a slow conformational change of ICL2. Moving one labeling site from T180 to R182, thus away from ICL2, depletes the sensitivity towards local motions of ICL2. We attribute the discrepancy between smFRET and DEER to the long-linker fluorophores that may amplify the rotational conformation change and/or local conformational change to a linear distance change compared to the short spin labels (Extended Data Fig. 16).'

Reviewer #2, point 4: This is related to point 3 above. Also for DEER two label pairs have to resolve the conformational change. I understand that the production of the protein is difficult but that is why I am fine with 2 sites for FRET but these sites have to be corroborate with two sites for DEER, especially if there is a discrepancy between the methods for one pair. I am sorry that I was a bit unclear on this point in the previous review.

At the beginning of this study, we screened several labeling pairs for DEER analysis. Besides the TM4-TM6 labeling pair, we also spin-labeled μ OR Δ 7 at the intracellular ends of TM1 (R95C) and TM6 (R276C). The distance distributions between TM1 and TM6 are shown in Figure R2. In contrast to the TM4-TM6 pair in the manuscript, which shows four conformational states, the TM1-TM6 pair only shows two conformational states in 36.4 Å and 44.5 Å, close to the predicted distances of 37.9 Å in the inactive and 45.3 Å in the G protein-coupled structure. No significant difference between Lofentanil and Lofentanil+Gi is observed. The reduced number of conformational states observed by the TM1-TM6 labeling pair is likely because TM6 moves on an arc when observed from TM1, resulting in similar distances (cf. [10.1016/j.cell.2018.12.005](https://doi.org/10.1016/j.cell.2018.12.005)). Further, for this construct, the labeling efficiency was strongly reduced leading to a weaker DEER signal and dramatically increased data acquisition time. The reduced signal also entails decreased spatial resolution and wider confidence bands, especially in the long distance range (active state).

Aim of this study was to capture the conformational ensemble of TM6 in μ OR and to identify important structure/function relationships. To be able to characterize TM6 motion in more detail, we focused on the TM4-TM6 pair for the DEER study. Nevertheless, the TM1-TM6 pair shows consistent results with the TM4-TM6 pair insofar that the Apo state exhibits a minor population of active conformation, while the super-efficacy agonist Lofentanil quantitatively stabilizes the active conformation.

Figure R2. DEER distance distributions of $\mu\text{OR}\Delta 7\text{-R95C/R276C}$ labeled by HO-1427. $\mu\text{OR}\Delta 7$ was labeled by HO-1427 in the intracellular ends of TM1 (R95C) and TM6 (R276C). **(A)** Background corrected dipolar evolutions and associated fits for the inactive (Apo), active (Lofentaniil), and G protein bound (Lof+Gi) states. **(B-C)** Corresponding distance distributions demonstrating receptor activation as reflected by TM6 outward movement also observed in high-resolution structures of mOR.

Reviewer #2, E: The authors are to be commended for their repeats and the large amount of work they had to put into it. I am fully satisfied with regard to this point. Having said that and without implication for this manuscript: I disagree with their reply that there is no standard in the field. The white paper of the DEER community does clearly state that repeats (in form of a combination of technical and biological repeats) are a must.

We thank the reviewer's comments and apologize for the misunderstanding. While we agree that the white paper states the requirement of technical and biological repeats, there is no standard about the exact number of repeats, which conditions, etc.

Reviewer 2, H, point 2: I see, well done.

We thank reviewer #2 for his time and effort spent on reviewing our manuscript.

Referee #3 (Remarks to the Author):

I thank the authors for their responses. The additions made to the manuscript, together with their point-by-point response, are much appreciated. I am satisfied with the corrections.

We thank reviewer #3 for the time and effort spent on the review.

Reviewer Reports on the Second Revision:

Referees' comments:

Referee #2 (Remarks to the Author):

The authors have done everything asked for.

I am convinced with regards to the linker unit.

The data from the new mutant are also convincing and support the conclusions.

A very nice and well done paper that I hope will be published.